# On Group Sufficiency Under Label Bias

**Haoran Zhang**    **Olawale Salaudeen**    **Marzyeh Ghassemi**
Massachusetts Institute of Technology
{haoranz, olawale, mghassem}@mit.edu

## Abstract

Real-world classification datasets often contain label bias, where observed labels differ systematically from the true labels at different rates for different demographic groups. Machine learning models trained on such datasets may then exhibit disparities in predictive performance across these groups. In this work, we characterize the problem of learning fair classification models with respect to the underlying ground truth labels when given only label biased data. We focus on the particular fairness definition of group sufficiency, i.e. equal calibration of risk scores across protected groups. We theoretically show that enforcing fairness with respect to label biased data necessarily results in group miscalibration with respect to the true labels. We then propose a regularizer which minimizes an upper bound on the sufficiency gap by penalizing a conditional mutual information term. Across experiments on eight tabular, image, and text datasets with both synthetic and real label noise, we find that our method reduces the sufficiency gap by up to 7.2% with no significant decrease in overall accuracy.

## 1 Introduction

Machine learning systems are increasingly being deployed in safety critical settings, from parole recommendations to medical triage and financial lending [1, 2, 3]. However, prior work has shown that such systems often exhibit biases in the form of performance discrepancies between demographic groups [4, 5, 6]. There exists a range of work which propose algorithmic approaches to correct these disparities, spanning from pre-processing [7, 8, 9], to in-processing [10, 11, 12, 13], to post-processing [14, 15, 16]. However, these methods rarely consider the *source* of these biases. Prior work has shown that naively enforcing these fairness definitions can have unintended side effects, leading to worse performance for all [17, 18, 19, 20], and exacerbating other types of biases [21, 22, 16].

In this work, we take a different approach and tackle fair classification from a data-centric perspective [23], focusing on datasets that contain significant differences in label errors between protected groups, also known as *label bias* [24]. For example, in the medical diagnosis setting, women with heart disease are underdiagnosed at higher rates than men with heart disease, and are instead more likely to be misdiagnosed with mental health conditions [25]. Similarly, Black patients are less likely to be prescribed opioid painkillers than White patients in emergency departments, even after adjusting for level of pain [26]. Finally, when screening resumes for job applications, applicants with African American names are much less likely to receive an interview than those with White names [27].

When downstream models are trained on these label-biased datasets (i.e. to diagnose, prescribe, or hire), the model may learn to predict noisy labels and propagate disparities in the historical data [28, 29], leading to the observed performance disparities. When the training dataset contains label bias, our goal is to learn a fair and performant model with respect to the true unobserved labels.

Learning under label noise is a well-studied problem [30, 31, 32, 33, 34], where methods aim to learn the best model with respect to the true labels, without any consideration for per-group performance (i.e. without considering label bias). Though there has been limited work looking at fairness under

39th Conference on Neural Information Processing Systems (NeurIPS 2025).

Figure 1: Our Problem Setup. (a) Observed labels are corrupted from unobserved true labels at different rates for each gender, which may be caused by misdiagnoses [25] or noisy automated processes used to extract labels [42]. (b) Training a model on these noisy labels and evaluating calibration curves with respect to true labels results in different calibration curves for each group. Specifically, each group's calibration curve differs from the marginal curve. We propose a method to achieve group sufficiency (zero sufficiency gap for all groups) under these conditions.

label bias [35, 36, 37], these works all focus on the fairness definitions of demographic parity and equal odds [15].

Here, we study the problem of learning classification models under label bias which satisfy a different fairness property — *group sufficiency*, which corresponds to equal calibration of the risk score across groups [21, 38]. A group sufficient score can then be post-processed to achieve perfect calibration for all groups via group-agnostic post-processing [21]. Importantly, group sufficiency is incompatible with demographic parity and equal odds on the risk score under different group base rates, so it is not possible to achieve all fairness definitions simultaneously [22]. Here, we focus on group sufficiency, as prior work have shown that it is a requirement to prevent discrimination or maximize utility in many contexts [39, 40, 41]. For example, Loi and Heitz [40] argue, from a moral philosophy perspective, that risk scores being miscalibrated for a particular demographic group may constitute discrimination against that group.

We start by theoretically characterizing how class-attribute conditioned label noise distorts per-group calibration and derive necessary and sufficient conditions for group insufficiency under label bias (Section 4). This demonstrates that naive models trained on noisy labeled data are necessarily *unfair* with respect to the true underlying labels, and motivates the need for specialized methods. Building on this theory, we propose a regularizer term, CMI-REG, based on penalizing conditional mutual information (Section 5). CMI-REG can be added to any existing off-the-shelf noise-robust loss. We evaluate CMI-REG with 7 base losses on 8 datasets spanning tabular, text, and image domains, finding that we effectively reduce the sufficiency gap while maintaining, and often improving, overall performance over the base losses. To the best of our knowledge, our work is the first to study this problem setting and propose a method to achieve this fairness definition.

**Contributions.** Our work makes the following contributions[1]:

1. We theoretically establish necessary and sufficient conditions for which label bias causes a lack of group sufficiency under class-attribute conditioned noise.

2. We propose a method to achieve sufficiency under label bias, assuming class-attribute conditioned noise. We theoretically prove that our method minimizes an upper bound on the sufficiency gap. Our method is a regularizer term that can be flexibly added to any existing loss-based noisy label learning method.

3. We empirically evaluate our method on 8 datasets with both synthetic and real-world noise, showing that it achieves sufficiency with respect to the true labels without a significant decrease in accuracy over the baselines.

---

[1]Code: `https://github.com/MLforHealth/sufficiency_label_bias`

## 2 Related Work

### 2.1 Learning Under Label Noise

There has been a long line of work on the problem of learning classification models under label noise. Under uniform label noise, symmetric loss functions such as mean absolute error (MAE) are theoretically noise robust [43]. This then motivated a line of work which adapts MAE to have better optimization properties and empirical performance, including improved MAE [44], symmetric cross entropy [45], generalized cross entropy (GCE) [30], and generalized Jensen-Shannon divergence [46].

Under class conditioned noise (where the probability of observing a noisy label depends only on the true label), one direction focuses on estimating the class transition matrix. When the class transition matrix is known, a simple weighting approach is sufficient [47, 48]. Methods to estimate the transition matrix have been proposed, given a small dataset with clean labels [49, 50], anchor points for each class [51, 52], or by examining the transition matrix of an ERM model trained on noisy labels [53, 54]. Liu and Guo [31] proposes a peer loss which, in the binary case, only requires knowledge of the sum of the off-diagonal transition matrix elements.

Under instance dependent label noise, there are no theoretical guarantees in general. Methods that have done well empirically in this area include those based on training dynamics, e.g. area under the margin [55], DataMap [56], first-time k-epoch learning [57], early regularization [58], as well as methods based on training multiple simultaneous models such as Co-teaching [59] and DivideMix [32].

However, given that all methods in this category are agnostic of the group attribute and do not explicitly enforce a fairness condition, they do not explicitly satisfy any fairness properties, and thus do not succeed in our problem setup as we will later show empirically.

### 2.2 Sufficiency as a Fairness Definition

Sufficiency of the risk score is a commonly used group fairness definition in the classification setting, which corresponds to equal calibration curves for all groups. It is a slightly weaker definition than calibration by group (i.e. all groups being perfectly calibrated) [21]. Prior work has shown that under different per-group prevalences of the label, it is impossible to simultaneously satisfy sufficiency and probabilistic equal odds [22, 60]. Liu et al. [38] show that ERM models inherently prefer sufficiency, and that intervening to achieve probabilistic equal odds necessarily worsens this condition. Empirically, this has been observed in images [19, 61] and tabular data [62, 41]. Raghavan [39] discuss the implications of this impossibility, making a case that the choice of fairness metric is highly context dependent. Similarly, Loi and Heitz [40] discuss, from a philosophical view, when group calibration is a required condition to avoid discrimination.

In order to achieve sufficiency, prior methods have been proposed based on mutual information [63] and multi-objective optimization [64, 65]. One related concept is multicalibration [66], which seeks to achieve approximate calibration for all computationally identifiable subgroups. However, multicalibration has been found to be far less sample efficient than standard post-hoc calibration methods [67]. Both multicalibration (when applied to provided attributes) and standard post-hoc calibration methods require access to the group attribute during inference, which our method will not assume. Regardless, no prior work has studied the setting of group sufficiency under label noise, which is the focus of this work.

### 2.3 Fairness Under Label Bias

Prior works on fairness under label bias have only focused on demographic parity and equal odds. Wick et al. [68] use simulated data to show that enforcing statistical parity on label-biased data may already achieve the same fairness definition on clean data, with increased accuracy. Wang et al. [35] study the problem of equal odds under class-attribute conditioned label noise, and propose two solutions based on an alternate constraint using a surrogate loss, and a group-weighted peer loss [31]. Zhang et al. [37] learn fair representations under class-conditional noise for the same definitions. Wu et al. [36] study fairness under instance-dependent noise but targets counterfactual fairness [69], which has been shown to have similar drawbacks as equal odds [70]. None of these works consider group sufficiency, nor do they provide guarantees for calibration error under noise.

Finally, Mhasawade et al. [24] propose various candidate causal graphs to model the label bias scenario and derive cases when sufficiency is satisfied without further intervention, but do not propose any algorithmic solutions in the remaining cases. Under their framework, we study the $Y \rightarrow Y_i$ and $Y_i \not\perp\!\!\!\perp A$ setting, under which they provide no guarantees. In contrast, we provide a theoretical characterization of this setting and propose a method to achieve group sufficiency under label bias.

# 3 Preliminaries

Let $X \in \mathcal{X}$ denote an input feature, $Y \in \{1, 2, \ldots, K\}$ the true (clean) label, and $\tilde{Y} \in \{1, 2, \ldots, K\}$ the observed (noisy) label. Additionally, let $G \in \mathcal{G}$ represent a group attribute. We have a dataset $\mathcal{D} = \{(x_i, g_i, y_i, \tilde{y}_i)\}_{i=1}^N$ with $(x_i, g_i, y_i, \tilde{y}_i) \sim P_{X,G,Y,\tilde{Y}}$, where the true labels $y_i$ are unobserved, and we only have access to the noisy labels $\tilde{y}_i$.

In practice, we want classifiers whose predicted probabilities accurately reflect true real-world outcomes, both overall and consistently across different demographic or group attributes. To quantify how well a model achieves these goals, we introduce the standard definition of group sufficiency, which quantifies whether predictions provide equal information about outcomes, regardless of group membership.

**Definition 3.1** (Group Sufficiency). A multiclass classifier $f : \mathcal{X} \rightarrow \Delta^{K-1}$ is *group sufficient* with respect to a label $Y$ and group $G$ if it satisfies:

$$Y \perp\!\!\!\perp G \mid f(X).$$

This condition implies that the prediction $f(X)$ captures information about $Y$ that renders the group attribute $G$ irrelevant given the model's output.

**Definition 3.2** (Group Sufficiency Gap). The per-group sufficiency gap, as well as the overall sufficiency gap for a model $f$, are defined as [38]:

$$\mathrm{Suf}_g(f) = \left( \sum_{k=1}^K \frac{1}{K} \mathbb{E}[|\mathbb{E}[\mathbf{1}_{Y=k} \mid f(X)_k] - \mathbb{E}[\mathbf{1}_{Y=k} \mid f(X)_k, G = g]|^p] \right)^{1/p}$$

$$\mathrm{Suf}(f) = \mathbb{E}_g[\mathrm{Suf}_g(f)]$$

Similarly, let $\widetilde{\mathrm{Suf}}$ denote the sufficiency gap with respect to $\tilde{Y}$. The *group sufficiency gap* quantifies the extent to which a model's predictions provide unequal information about the true label $Y$ across different groups $G$; it measures deviations from the ideal group sufficiency condition $Y \perp\!\!\!\perp G \mid f(X)$. This gap is zero when $f(X)$ is group sufficient—Lemma A.1.

To formally analyze the effect of label noise and group membership on calibration and sufficiency, we introduce explicit assumptions about the data-generating process. These assumptions clarify the relationship between noisy and true labels, conditional independence, and the structure of the noise across different groups.

**Assumption 3.1** (Class-Attribute Conditioned Noise). The noisy label $\tilde{Y}$ is conditionally independent of the input $X$ given the true label $Y$ and group attribute $G$:

$$\tilde{Y} \perp\!\!\!\perp X \mid (Y, G).$$

**Definition 3.3** (Group-Conditional Transition Matrix). For each group $g \in \mathcal{G}$, define a row-stochastic noise transition matrix $T^g \in [0,1]^{K \times K}$ where

$$T_{i,j}^g = \mathbb{P}(\tilde{Y} = j \mid Y = i, G = g).$$

**Assumption 3.2** (Invertible Group-Conditional Transition Matrix). $T^g$ is invertible for all groups $(g \in \mathcal{G})$.

These assumptions are standard in the literature on learning under label noise [48, 71]. Importantly, the set of singular matrices has Lebesgue measure zero within the space of all row-stochastic matrices [72], and strictly diagonally dominant matrices—which naturally arise when noise rates are sufficiently small—are always invertible [73].

**Definition 3.4** (Label Bias). Let $T \in [0,1]^{K \times K}$ where $T_{i,j} = P(\tilde{Y} = j \mid Y = i) = \sum_{g \in \mathcal{G}} T_{i,j}^g P(G = g)$. If $\exists g \in \mathcal{G}$ such that $T^g \neq T$, then we say that the observed labels $\tilde{Y}$ are *label biased* relative to $Y$.

# 4 Main Result and Problem Statement

First, we characterize the necessary and sufficient conditions for which we would expect a non-zero sufficiency gap.

**Theorem 4.1** (Closed-Form Decomposition of the Sufficiency Gap). Let $s = f(X) \in \Delta^{K-1}$ be the model score, $\tilde{\pi}_g(s) = P(\tilde{Y} = \cdot \mid f(X) = s, G = g)$ the group-conditional noisy label posterior, and $\tilde{\pi}(s) = P(\tilde{Y} = \cdot \mid f(X) = s)$ the marginal counterpart. Then under Assumptions 3.1 and 3.2, for any $p \geq 1$,

$$\mathrm{Suf}_g(f) = \left( \frac{1}{K} \mathbb{E} \left[ \left\| \underbrace{((T^g)^\top)^{-1} \Delta_{\tilde{\pi}}(f(X))}_{\text{posterior shift}} + \underbrace{\tilde{\pi}(f(X)) \Delta_{T^g}}_{\text{noise matrix mismatch}} \right\|_p^p \right] \right)^{1/p}, \tag{1}$$

where $\Delta_{\tilde{\pi}}(f(X)) = (\tilde{\pi}_g(f(X)) - \tilde{\pi}(f(X)))$ and $\Delta_{T^g} = (((T^g)^\top)^{-1} - (T^\top)^{-1})$.

**Corollary 4.1.1** (Necessary Condition for Sufficiency Gap). Under Assumptions 3.1 and 3.2, suppose that $f$ satisfies group sufficiency with respect to noisy labels (i.e. $\tilde{Y} \perp\!\!\!\perp G \mid f(X)$). Then, if $\exists g \in \mathcal{G}$ such that $\mathrm{Suf}_g(f) > 0$, it must be that $T^g \neq T$.

Thus, when $f$ is fair with respect to the noisy labels (which ERM models trained on noisy data have a tendency to be, as the sufficiency gap is upper bounded by its excess risk [38]), label bias is *necessary* for a sufficient gap. The contrapositive of this corollary also shows that under uniform label noise which is group agnostic, a group sufficient classifier with respect to noisy labels (implying $\widetilde{\mathrm{Suf}}(f) = 0$) will also be group sufficient to the true labels. This implies that if we know apriori that the noise rates do not differ across groups (i.e. there is no label bias), enforcing group sufficiency with respect to the noisy labels is a valid mitigation strategy. Thus, for the rest of this work, we will deal with the case of non-group-agnostic noise.

**Remark 4.2** (Sufficient Condition for Sufficiency Gap). Theorem 4.1 implies that under label bias (i.e. $T^g \neq T$) and a non-trivial classifier ($f(X) \neq 0$), we have that $\mathrm{Suf}_g(f) > 0$, so long as the model is group-sufficient for the noisy labels ($\Delta_{\tilde{\pi}}(f(X)) = 0$) and $\tilde{\pi}(f(X))$ is not in the null space of $\Delta_{T^g}$. Specifically, even though $T$ and $T^g$ are different, if the marginal $\tilde{\pi}$ does not have sufficient mass in the directions that they are different, then the sufficiency gap is unaffected since the inverses of $T$'s project $\tilde{\pi}$ to the same output and cancel out, resulting in $\mathrm{Suf}_g(f) = 0$.

We define and provide similar bounds for the per-group calibration error in Appendix A. Further, we characterize the behavior of $\mathrm{Suf}_g$ under perturbations to the transition matrix $T^g$, which may happen when $T^g$ is estimated from observational data. All proofs can be found in Appendix A.

**Problem Statement.** Our goal is to learn a risk score $f_\theta : \mathcal{X} \to \Delta^{K-1}$ satisfying Definition 3.1. Specifically, given a loss function $l((x, y), f) \to \mathbb{R}$. We would additionally like to achieve:

$$\min_{\theta \in \Theta} \mathbb{E}_{(X,Y) \sim p_{X,Y}} [l((X, Y), f_\theta)]$$
$$s.t. \quad Y \perp\!\!\!\perp G \mid f_\theta(X) \tag{2}$$

# 5 Methodology

To guide the design of our method, we derive an upper bound on the group sufficiency gap in terms of the conditional mutual information between the label and group, given the model's prediction. This result highlights that reducing the dependence between $Y$ and $G$ conditioned on $f(X)$ is sufficient to control the sufficiency gap:

**Proposition 5.1** (Conditional Mutual Information Bound on Sufficiency Gap).

$$\mathrm{Suf}_g(f) \leq \frac{\sqrt{2}}{K} \sqrt{\frac{I(Y; G \mid f(X))}{P(G = g)}}; \qquad \mathrm{Suf}(f) \leq \frac{\sqrt{2}}{K} \sqrt{I(Y; G \mid f(X))}.$$

This bound motivates our core objective: to minimize the sufficiency gap by minimizing the conditional mutual information $I(Y; G \mid f(X))$. However, since the true labels $Y$ are unobserved in

practice, directly estimating this quantity is non-trivial. In the following section, we describe how we approximate this objective using the noisy labels $\tilde{Y}$. Formal proofs are provided in Appendix A.

Under Assumptions 3.1 and 3.2, we propose a differentiable regularizer that approximates and minimizes the conditional mutual information upper bound using the observed noisy labels.

$$\min_\theta \sum_{i=1}^n \ell_{pred}(f_\theta, x_i, \tilde{y}_i) + \lambda R(f_\theta), \tag{3}$$

where $f_\theta : \mathcal{X} \to \mathcal{Y}$, $\lambda > 0$, and $\ell_{pred}$ is a pointwise loss function, potentially selected for robustness to label noise, e.g., Peer Loss [31] or GCE [30].

**CMI Regularizer**  We can decompose $I(Y; G \mid f(X))$ into the difference of two entropy terms:

$$I(Y; G \mid f(X)) = H(Y \mid f_\theta(X)) - H(Y \mid f_\theta(X), G)$$
$$= \mathbb{E}_{X,Y}[-\log \mathbb{P}(Y \mid f_\theta(X))] - \mathbb{E}_{X,Y,G}[-\log \mathbb{P}(Y \mid f_\theta(X), G)]$$

For a mini-batch $\mathcal{B}$ of size $B$, each entropy term can be estimated using a Monte Carlo estimator:

$$\widehat{H}(Y \mid f_\theta(X), G) = \frac{1}{B} \sum_{i \in \mathcal{B}} (-\log \mathbb{P}(Y = y_i \mid f_\theta(X) = f_\theta(x_i), G = g_i))$$

$$= \frac{1}{B} \sum_{i \in \mathcal{B}} (-\log \mathbf{p}_Y(\cdot \mid f_\theta(X) = f_\theta(x_i), G = g_i))_{y_i}$$

where $\mathbf{p}_Y(\cdot \mid f_\theta(X) = f_\theta(x_i), G = g_i) := [\mathbb{P}(Y = 1 \mid f_\theta(X) = f_\theta(x_i), G = g_i), ..., \mathbb{P}(Y = K \mid f_\theta(X) = f_\theta(x_i), G = g_i)]$, and $(\cdot)_{\tilde{y}_i}$ denotes taking the $\tilde{y}_i$-th element of the $K$-dimensional vector.

As we do not observe $y_i$, we can apply the backward correction [48] to this estimator, where log is applied elementwise:

$$\widehat{H}(Y \mid f_\theta(X), G) = \frac{1}{B} \sum_{i \in \mathcal{B}} \left( -T_{g_i}^{-1} \log \mathbf{p}_Y(\cdot \mid f_\theta(X) = f_\theta(x_i), G = g_i) \right)_{\tilde{y}_i} \tag{4}$$

and similarly

$$\widehat{H}(Y \mid f_\theta(X)) = \frac{1}{B} \sum_{i \in \mathcal{B}} \left( -T_{g_i}^{-1} \log \mathbf{p}_Y(\cdot \mid f_\theta(X) = f_\theta(x_i)) \right)_{\tilde{y}_i} \tag{5}$$

Note that the population entropy $H(Y \mid f(X))$ does not depend on $G$. However, as we only observe $\tilde{Y}$, our estimator must invert the noise process that generated $\tilde{Y}$. Under Assumption 3.1, the unbiased backward-correction of $Y$ requires the group-specific matrix $T_{g_i}^{-1}$ for the $i$-th sample. Using the marginal matrix $T^{-1}$ would bias the estimator when there is label bias.

**Heads for Entropy Estimation**  To estimate $\mathbf{p}_Y$ given only noisy data, we parameterize two functions $h_\phi : \Delta^{K-1} \to \Delta^{K-1}$ and $h_\psi : \Delta^{K-1} \times \mathcal{G} \to \Delta^{K-1}$ each by a single linear layer, which we then learn by using a standard cross entropy loss with the forward correction [48]:

$$h_\phi = \arg\min_\phi \frac{1}{B} \sum_{i \in \mathcal{B}} -\log(T_{g_i}^\top h_\phi(f_\theta(x_i)))_{\tilde{y}_i}, \tag{6}$$

$$h_\psi = \arg\min_\psi \frac{1}{B} \sum_{i \in \mathcal{B}} -\log(T_{g_i}^\top h_\psi(f_\theta(x_i), g_i))_{\tilde{y}_i} \tag{7}$$

Thus, we have that

$$R(f_\theta) = \hat{I}(Y; G \mid f(X)) = \frac{1}{B} \sum_{i \in \mathcal{B}} \left( -T_{g_i}^{-1} \log h_\phi(f_\theta(x_i)) \right)_{\tilde{y}_i} - \left( -T_{g_i}^{-1} \log h_\psi(f_\theta(x_i), g_i) \right)_{\tilde{y}_i} \tag{8}$$

A single linear layer is each used to parameterize $h_\phi$ and $h_\psi$, as similar models are frequently learned for post-hoc calibration [74]. In practice, $h_\phi$ and $h_\psi$ operate on the logits of $f$ instead of the output

---

**Algorithm 1:** Single Step Update of CMI-REG

---

**Input:** Batch $\mathcal{B} = \{(\mathbf{x}_i, \mathbf{g}_i, \tilde{\mathbf{y}}_i)\}_{i=1}^{B}$ ;
Per–group transition matrices $\{T^g \in [0,1]^{K \times K}\}_{g \in \mathcal{G}}, \lambda > 0, C \geq 1$;
Learning rates $\alpha_f, \alpha_h > 0$;
Current classifier and auxiliary heads: $f_\theta, h_\phi, h_\psi$ ;
$\ell_{pred}(z_i, \tilde{y}_i)$;
**Output:** Updated $f_\theta, h_\phi, h_\psi$

**1** $z_i \leftarrow f_\theta(x_i) \ \forall i \in \mathcal{B}$ ;  // logits
**2 for** $j = 1$ **to** $C$ **do**  // update heads, freeze $\theta$

**3** $\quad \phi \leftarrow \phi - \alpha_h \nabla_\phi \dfrac{1}{B} \sum_{i \in \mathcal{B}} -\log \left(T_{g_i}^\top h_\phi(z_i)\right)_{\tilde{y}_i}$ ;  // Eq. (6)

**4** $\quad \psi \leftarrow \psi - \alpha_h \nabla_\psi \dfrac{1}{B} \sum_{i \in \mathcal{B}} -\log \left(T_{g_i}^\top h_\psi(z_i, g_i)\right)_{\tilde{y}_i}$ ;  // Eq. (7)

**5** $\mathcal{L}_{\text{pred}} = \dfrac{1}{B} \sum_{i \in \mathcal{B}} \ell_{\text{pred}}(z_i, \tilde{y}_i)$;
**6** Compute $R_B$ using Equation (8);
**7** $\mathcal{L} = \mathcal{L}_{\text{pred}} + \lambda R_B$
**8** $\theta \leftarrow \theta - \alpha_f \nabla_\theta \mathcal{L}$;
**9 return** $f_\theta, h_\phi, h_\psi$;

---

probabilities. Given a batch $B$, we first update $h_\phi$ and $h_\psi$ using gradient descent on Equations (6) and (7) for $C$ iterations, and then update $f_\theta$ using Equation (3) once. As the backward corrected entropy estimators (Equations 4 and 5) can produce entropies outside of $[0, \log K]$, we clip estimated entropies to this range. The full algorithm can be found in Algorithm 1. When the two heads are perfectly learned, $R(f_\theta)$ is an unbiased estimator of the conditional mutual information (see Lemma A.5).

**Directly Regularizing Sufficiency (SUF-REG)**  In parallel, we consider a direct penalty on the sufficiency gap by plugging in estimates of $\tilde{\pi}$ and $\tilde{\pi}_g$ into Theorem 4.1. Concretely, $\tilde{\pi}$ and $\tilde{\pi}_g$ are estimated by two linear heads and trained in an alternating manner similar to CMI-REG. We refer to this regularizer term as SUF-REG, and it can be combined with any $\ell_{pred}$ similar to CMI-REG. We emphasize that SUF-REG is also a novel contribution that has not been explored in prior work. This variant avoids mutual-information estimation but can be higher-variance. An algorithm outline for SUF-REG can be found in Appendix C.

## 6   Experiments

**Datasets and Noise Types**  We evaluate our method using the 8 datasets shown in Table 1, which span tabular, image, and text modalities, and a range of number of classes. For datasets with synthetic noise, we experiment with two synthetic noise types: *Group Uniform* (where uniform noise is added only for one group), and *Group Asymmetric* (where class-specific noise is only added for one group). Additional details of how the per-group transition matrices are constructed can be found in Appendix B.2. Datasets with real noise have label noise derived from multiple annotators (in the case of `civilcomments` and `cifar10ns`) or from human annotators of noisy web data (for `clothing1m`). Further details of dataset construction can be found in Appendix B.1.

**Training**  We train MLP, ImageNet-pretrained ResNet-18 [75], and BERT-base [76] models for tabular, image, and text modalities respectively, optimized using the Adam optimizer [77]. All datasets are divided into 60%/20%/20% training/validation/test splits. For synthetic noise, to mitigate the confounding effect of different marginal noise rates depending on group proportions, we balance the groups by subsampling the majority group during both training and testing. Hyperparameters were selected using a random hyperparameter search [78] with 20 runs. For the full hyperparameter grid for each method, see Appendix B.3. Each hyperparameter setting was repeated three times with different random seeds, which affects the dataset split, model initialization, and random noise (in the case of synthetic noise).

Table 1: Datasets used this paper. All attributes contain two groups. $n$: number of samples, $K$: number of classes. Data processing details can be found in Appendix B.1.

| Dataset | Modality | Task | Attribute | $n$ | $K$ | Noise Type |
|---|---|---|---|---|---|---|
| adult [80] | Tabular | Income $\geq$ 50k | Gender | 32,561 | 2 | Synthetic |
| lsac [81] | Tabular | Student passes the bar | Race | 18,337 | 2 | Synthetic |
| crime [82, 83] | Tabular | Binned rate of violent crime | Primary ethnicity | 1,994 | 5 | Synthetic |
| income [84] | Tabular | Binned income | Race | 1,445,699 | 3 | Synthetic |
| grades [85] | Tabular | Student passes exam | Gender | 856 | 2 | Real |
| civilcomments [86] | Text | Comment is toxic | Contains identity | 448,000 | 2 | Real |
| clothing1m [87] | Image | Type of clothing | Contains face | 1,072,409 | 14 | Real |
| cifar10ns [88, 89] | Image | Image classification | Image is grayscale | 60,000 | 10 | Real |

To select the optimal hyperparameters, we experiment with three model selection strategies: (1) *Noisy Brier*: We select the model with the lowest overall Brier score on the noisy validation set. The Brier score is chosen as the selection criteria as it accounts for both performance and calibration. (2) *Theory*: We select the model with the smallest value from Theorem 4.1 on the noisy validation set, which requires knowledge of the transition matrices but not any clean labels, (3) *Clean Brier*: We select the model with the lowest overall Brier score on the *clean* validation set, which requires a validation set with clean labels. For our method (and baselines) which require transition matrices, we experiment with two settings: (1) *Known*: we use the true $T^g$, (2) *Estimated*: we estimate $T^g$ using Confident Learning [54].

**Baselines**  We consider the following off-the-shelf loss functions for $\ell_{pred}$: Cross Entropy (CE), Backward [48], Forward [48], Generalized Cross Entropy (GCE) [30], $\alpha-$weighted peer loss (Peer) [31], Early Learning Regularization (ELR) [58], and Determinant Mutual Information (DMI) [79]. We compare our methods (CMI-REG and SUF-REG) against each of the baseline losses. We also compare against the group-weighted peer loss for achieving equal odds from Wang et al. [35] ("GroupPeer"). Note that this method is only applicable when $K = 2$. We additionally compare against applying multicalibration ("MC") post-processing using Algorithm 1 from Hébert-Johnson et al. [66] (specifically, using the implementation from Hansen et al. [67]), using the single attribute to define $\mathcal{C}$, and defining the calibration set to be a 40% subset of the original training set. This post-processing is done to the scores of an ERM model trained on labels produced by per-group stochastic flipping according to the row-normalized $(T_g)^{-1}$. Finally, as oracle, we train an ERM model on the clean training data.

**Metrics**  All metrics shown are evaluated on the cleanly labeled test set. To evaluate overall performance, we use the one-vs-rest AUROC macro-averaged across classes. To evaluate fairness, we compute the sufficiency gap $\mathrm{Suf}(f)$ as in Definition 3.2 with $p = 1$, by using a binning estimator with 10 equally sized bins.

## 7  Results

We present results for the four tabular datasets with synthetic group asymmetric noise, clean Brier selection, and known transition matrices in the main paper. Additional experimental results for group uniform noise, real-world noise, other selection criteria, estimated transition matrices, and additional synthetic noise levels can be found in Appendix D.

**On Performance and Fairness**  In Table 2, we show the overall performance and sufficiency gap of SUF-REG and CMI-REG when added to seven different base losses. We find that in the large majority of cases, the addition of either regularizer results in a lower sufficiency gap, with CMI-REG exhibiting marginally better fairness. However, the confidence intervals are generally large as calibration error estimation has high variance [90]. Regardless, the model which achieves the lowest sufficiency gap is consistently regularized by CMI-REG. The base losses which function best with these regularization terms are GCE and Peer loss. Interestingly, in several cases, adding the regularizer term actually improves overall performance. Similar results for group uniform noise and real-world noise can be found in Appendix D. When the estimated transition matrix is used instead (Appendix D.5), we find that regularizers still generally improve fairness, though the performance improvements become limited.

Table 2: Clean test-set AUROC and sufficiency gap (Suf) of our regularizers when added to off-the-shelf loss functions, for tabular datasets with synthetic group asymmetric noise and known transition matrices. Models are selected using the clean Brier score. The best method within each base loss is **bolded**, and the best model overall is also **underlined**.

| | income | | adult | | lsac | | crime | |
|---|---|---|---|---|---|---|---|---|
| | AUROC ↑ | Suf ↓ | AUROC ↑ | Suf ↓ | AUROC ↑ | Suf ↓ | AUROC ↑ | Suf ↓ |
| CE | 0.731 (0.000) | 0.040 (0.003) | 0.899 (0.000) | 0.029 (0.003) | 0.581 (0.049) | 0.108 (0.003) | 0.741 (0.025) | 0.075 (0.011) |
| CE + Suf-Reg | 0.732 (0.000) | 0.035 (0.007) | 0.894 (0.000) | 0.029 (0.004) | **0.712** (0.022) | 0.089 (0.007) | **0.804** (0.021) | **0.057** (0.015) |
| CE + CMI-Reg | **0.736** (0.001) | **0.033** (0.001) | **0.903** (0.003) | **0.016** (0.004) | 0.708 (0.077) | **0.072** (0.026) | 0.748 (0.030) | 0.065 (0.007) |
| Forward | 0.801 (0.002) | 0.038 (0.002) | 0.887 (0.001) | 0.024 (0.004) | 0.436 (0.006) | 0.097 (0.001) | 0.788 (0.011) | 0.056 (0.010) |
| Forward + Suf-Reg | **0.812** (0.002) | 0.059 (0.003) | **0.898** (0.000) | 0.032 (0.001) | **0.690** (0.011) | 0.091 (0.016) | 0.789 (0.006) | 0.053 (0.006) |
| Forward + CMI-Reg | 0.802 (0.003) | **0.019** (0.003) | 0.887 (0.001) | **0.023** (0.006) | 0.664 (0.247) | **0.075** (0.052) | **0.797** (0.000) | **0.052** (0.016) |
| Backward | 0.794 (0.002) | 0.042 (0.002) | 0.887 (0.001) | 0.021 (0.003) | 0.473 (0.042) | 0.100 (0.003) | 0.711 (0.007) | 0.080 (0.014) |
| Backward + Suf-Reg | **0.800** (0.002) | **0.032** (0.002) | **0.888** (0.001) | 0.025 (0.004) | **0.631** (0.028) | 0.097 (0.017) | 0.721 (0.023) | 0.073 (0.002) |
| Backward + CMI-Reg | 0.797 (0.003) | 0.036 (0.005) | **0.888** (0.001) | **0.018** (0.007) | 0.452 (0.032) | **0.091** (0.009) | **0.739** (0.065) | **0.066** (0.005) |
| DMI | 0.590 (0.115) | 0.047 (0.016) | **0.887** (0.001) | 0.029 (0.004) | 0.476 (0.184) | **0.088** (0.000) | 0.406 (0.033) | **0.087** (0.033) |
| DMI + Suf-Reg | **0.591** (0.111) | **0.045** (0.016) | 0.877 (0.002) | 0.028 (0.006) | 0.478 (0.167) | 0.090 (0.000) | 0.395 (0.033) | 0.088 (0.033) |
| DMI + CMI-Reg | 0.590 (0.119) | 0.048 (0.017) | 0.885 (0.002) | **0.020** (0.006) | **0.478** (0.170) | 0.090 (0.001) | **0.484** (0.042) | 0.107 (0.023) |
| ELR | 0.731 (0.001) | 0.040 (0.003) | **0.898** (0.001) | 0.030 (0.005) | 0.364 (0.266) | **0.061** (0.050) | 0.662 (0.091) | 0.092 (0.012) |
| ELR + Suf-Reg | **0.754** (0.000) | **0.025** (0.003) | 0.873 (0.005) | 0.022 (0.002) | **0.465** (0.054) | 0.102 (0.001) | **0.781** (0.054) | **0.054** (0.011) |
| ELR + CMI-Reg | 0.738 (0.003) | 0.033 (0.002) | 0.860 (0.010) | **0.019** (0.001) | 0.319 (0.040) | 0.089 (0.014) | 0.664 (0.070) | 0.088 (0.001) |
| GCE | 0.730 (0.002) | 0.041 (0.004) | **0.894** (0.002) | **0.022** (0.001) | 0.462 (0.028) | 0.099 (0.002) | 0.739 (0.025) | 0.077 (0.014) |
| GCE + Suf-Reg | **0.748** (0.001) | 0.041 (0.002) | 0.884 (0.003) | 0.027 (0.000) | **0.590** (0.082) | 0.097 (0.004) | **0.771** (0.047) | **0.061** (0.008) |
| GCE + CMI-Reg | 0.735 (0.000) | **0.036** (0.002) | **0.894** (0.008) | **0.022** (0.002) | 0.257 (0.039) | **0.050** (0.018) | 0.771 (0.014) | 0.062 (0.000) |
| Peer | 0.730 (0.022) | **0.030** (0.001) | 0.899 (0.000) | 0.029 (0.004) | 0.553 (0.063) | 0.110 (0.003) | 0.684 (0.015) | **0.069** (0.003) |
| Peer + Suf-Reg | **0.732** (0.002) | 0.035 (0.005) | 0.890 (0.000) | 0.031 (0.003) | **0.590** (0.032) | 0.098 (0.001) | **0.734** (0.032) | 0.077 (0.000) |
| Peer + CMI-Reg | 0.686 (0.002) | 0.054 (0.003) | **0.901** (0.003) | **0.028** (0.002) | 0.551 (0.058) | **0.096** (0.006) | 0.630 (0.055) | 0.070 (0.021) |
| MC | 0.561 (0.019) | 0.078 (0.010) | 0.787 (0.004) | 0.017 (0.001) | 0.500 (0.000) | 0.107 (0.007) | 0.500 (0.000) | 0.104 (0.012) |
| GroupPeer | – | – | 0.898 (0.001) | 0.035 (0.004) | 0.664 (0.010) | 0.095 (0.023) | – | – |
| Oracle | 0.854 (0.001) | 0.003 (0.000) | 0.919 (0.002) | 0.013 (0.001) | 0.838 (0.011) | 0.018 (0.009) | 0.825 (0.013) | 0.046 (0.010) |

We find that naively adapting a post-hoc calibration method (multicalibration) results in generally worse performance and limited fairness gains, especially on `lsac` and `crime`. We hypothesize this stems from the limited effectiveness of the noise-aware label-flipping procedure used to construct training labels, and MC's sample-inefficient nature [67] combined with the smaller, class-imbalanced datasets.

**Pareto Fronts** In Figure 2, we show the Pareto front of AUROC vs. sufficiency gap for two datasets. We find that CMI-Reg and Suf-Reg Pareto-dominate the base losses in almost all cases. We also find that these two regularizers consistently achieve the best AUROC-sufficiency trade-off when comparing across all base losses (Appendix D.3).

**On Model Selection** In Appendix D.4, we examine the impact of varying the model selection criteria. As expected, selecting using clean Brier gives the best performance and lowest sufficiency gap. Selecting using the theoretical expression (Theorem 4.1) on the noisy validation set gives similarly low sufficiency gaps, but compromises significantly on performance. Selecting using the noisy Brier gives mediocre performance with slightly worse sufficiency. Across all selection criteria, selected models with regularization terms have lower sufficiency gaps than the base model in almost all cases.

## 8 Discussion

**On Performance and Fairness** Adding either CMI-Reg or Suf-Reg to seven noise-robust base losses rarely harmed AUROC and, in a surprising number of cases, improved AUROC, by up to 25.4% in one instance. We attribute this to the fact that both regularizers bias optimization toward decision boundaries that are locally calibrated with respect to the true labels across groups, and we speculate such decision boundaries are also more likely to coincide with high performing solutions with respect to the clean data.

**Comparing CMI-Reg and Suf-Reg** Even though CMI-Reg minimizes an upper bound while Suf-Reg minimizes the direct sufficiency gap, we observe that CMI-Reg performs better than Suf-Reg. Across eight combinations of datasets and noise types, CMI-Reg achieves the lowest sufficiency gap seven times, and Suf-Reg once. We attribute this to optimization and finite sample effects. First, Suf-Reg must backpropagate through an inverted $T$ inside of a $p$-norm, which may be

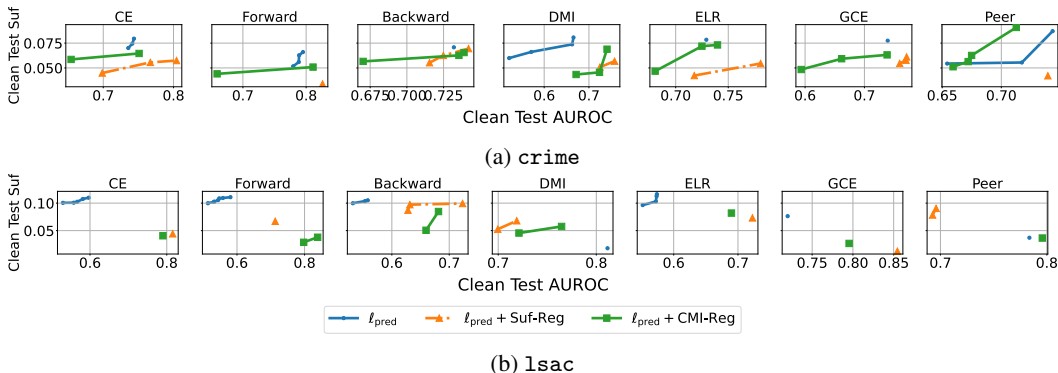

(a) `crime`

(b) `lsac`

Figure 2: Pareto fronts of AUROC versus Suf of CMI-REG, SUF-REG, and baseline loss for tabular datasets with synthetic group asymmetric noise and known transition matrices. A curve towards the bottom right of the plot is desirable as it achieves the best fairness-accuracy trade-off. We observe that CMI-REG and SUF-REG Pareto-dominate the base loss for most losses. Pareto curves for remaining datasets can be found in Appendix D.3.

more numerically challenging than simply backpropagating through an inverted $T$ multiplied by the log-likelihood, leading to abnormal gradient behavior. Second, CMI-REG consists of two convex cross-entropy terms, which may lead to a smoother optimization landscape than the piecewise-linear absolute error terms (when $p = 1$) inside SUF-REG.

## 9 Conclusion

In this work, we studied the problem of achieving group sufficiency under label bias. We have shown theoretically that under certain conditions, label bias is necessary and sufficient to induce a sufficiency gap. To address this, we introduced CMI-REG and SUF-REG, two drop-in regularizers that provably upper bound the sufficiency gap, and in practice, consistently reduce the sufficiency gap across eight datasets.

**Limitations** Our analysis rests on the class-attribute conditioned noise model (Assumption 3.1). When noise is instead instance-dependent [36, 91, 92], the regularizers SUF-REG and CMI-REG will become biased. Theoretically studying the effect of instance-dependent label bias is an area of future work. Next, our method assumes knowledge (or estimation) of the transition matrices $T^g$, and this estimation problem is impossible in general without further information or assumptions [93, 52]. These assumptions may be violated in real-world datasets, and applying our method using an imperfect transition matrix will result in worse performance as empirically observed. Finally, we measure fairness with respect to a single protected attribute at a time. Studying label bias between intersections of (potentially many) attributes is another area of future work.

## Acknowledgments and Disclosure of Funding

This research was supported in part by an Optum Labs Research Award, a National Science Foundation (NSF) 22-586 Faculty Early Career Development Award (#2339381), a Gordon & Betty Moore Foundation award, a Google Research Scholar award, and the AI2050 Program at Schmidt Sciences.

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

# A Proof of Lemmas and Theorems

## A.1 Definitions

**Definition A.1** (Calibration Error). The marginal calibration error of a multiclass classifier $f : \mathcal{X} \to \Delta^{K-1}$ with respect to Y and a $p$-norm ($p \geq 1$) is defined as [90, 94]:

$$\mathrm{CE}(f) = \left( \sum_{k=1}^{K} \frac{1}{K} \mathbb{E}[|f(X)_k - \mathbb{E}[\mathbf{1}_{Y=k} \mid f(X)_k]|^p] \right)^{1/p}$$

When $K = 2$ and $p = 1$, this is equivalent to the definition of Expected Calibration Error (ECE).

**Definition A.2** (Per-Group Calibration Error). Let $g$ denote a particular group:

$$\mathrm{CE}_g(f) = \left( \sum_{k=1}^{K} \frac{1}{K} \mathbb{E}[|f(X)_k - \mathbb{E}[\mathbf{1}_{Y=k} \mid f(X)_k, G = g] \mid G = g|^p] \right)^{1/p}$$

$\widetilde{\mathrm{CE}}(f)$ and $\widetilde{\mathrm{CE}}_g(f)$ are defined similarly with respect to $\tilde{Y}$.

## A.2 $\mathrm{Suf}_g(f)$ and Group Sufficiency

**Lemma A.1.** When $K = 2$, $f$ is group sufficient $\iff \forall g \in \mathcal{G} : \mathrm{Suf}_g(f) = 0$

*Proof.* ($\Longrightarrow$). Write $S := f(X) \in [0, 1]$. Because $K = 2$, the prediction is completely characterised by $S$: $P(Y = 1 \mid S) = S$, $P(Y = 0 \mid S) = 1 - S$.

Fix any group $g \in \mathcal{G}$ and any score value $s \in [0, 1]$ with $\Pr(S = s, G = g) > 0$. Group sufficiency gives the conditional independence

$$P(Y = y \mid S = s, G = g) = P(Y = y \mid S = s), \qquad y \in \{0, 1\}.$$

Hence

$$\mathbb{E}[\mathbf{1}_{Y=1} \mid S = s, G = g] = P(Y = 1 \mid S = s, G = g) = P(Y = 1 \mid S = s) = \mathbb{E}[\mathbf{1}_{Y=1} \mid S = s].$$

Substituting this equality into the definition of the sufficiency gap (Definition 3.2) yields $\mathrm{Suf}_g(f) = 0$.

($\Longleftarrow$) **for $K = 2$.** Assume $K = 2$ and that $\mathrm{Suf}_g(f) = 0$ for every group $g$. Let $s$ be any score value taken by $f(X)$ and let $g \in \mathcal{G}$ be fixed. Write $p_g(s) := P(Y = 1 \mid f(X) = s, G = g)$ and $p(s) := P(Y = 1 \mid f(X) = s)$. Because $\mathrm{Suf}_g(f) = 0$, the absolute difference in Definition 3.2 is zero, so $p_g(s) = p(s)$ almost surely. For binary labels, $P(Y = 0 \mid \cdot) = 1 - P(Y = 1 \mid \cdot)$, hence

$$P(Y = 0 \mid f(X) = s, G = g) = 1 - p_g(s) = 1 - p(s) = P(Y = 0 \mid f(X) = s),$$

so the conditional distributions of $Y$ given $f(X) = s$ are identical for every group. Therefore $P(Y = y, G = g \mid f(X) = s) = P(Y = y \mid f(X) = s) P(G = g \mid f(X) = s)$, i.e. $Y \perp\!\!\!\perp G \mid f(X)$, establishing group sufficiency.

$\square$

## A.3 Bound on CE

**Theorem A.2** (Lower Bound on $CE_g(f)$). Let $A^g = ((T^g)^\top)^{-1}$, and $s_k^g(X) = \sum_{k=1}^{K}(\mathbf{1}_{k=j} - A_{kj}^g)\mathbb{E}[\mathbf{1}_{\tilde{Y}=k} \mid f(X)_k, G = g]$. Then, under Assumption 3.1,

$$CE_g(f) \geq \left| \widetilde{\mathrm{CE}}_g(f) - \left( \frac{1}{K} \sum_{k=1}^{K} \mathbb{E}[|s_k^g(X)|^p \mid G = g] \right)^{1/p} \right|$$

For non-zero noise rate (e.g. $(T^g_{(0,1)}, T^g_{(1,0)}) \neq (0,0)$), this gives that $\mathrm{CE}_g(\tilde{f}^*_g(X)) > 0$, where $\tilde{f}^*_g$ is the Bayes optimal classifier with respect to noisy labels. Thus, learning the best model solely on the noisy labels will give us an uncalibrated model with respect to the true labels.

*Proof.* Fix a group $g \in \mathcal{G}$ and abbreviate all expectations and probabilities *conditionally* on $G = g$ (we omit the conditioning in the notation for clarity). For every $k \in \{1, \dots, K\}$ define

$$\tilde{\pi}_k(X) := \mathbb{E}[\,\mathbf{1}_{\tilde{Y}=k} \mid f(X)_k\,], \qquad r_k(X) := \mathbb{E}[\,\mathbf{1}_{Y=k} \mid f(X)_k\,].$$

Under Assumptions 3.1 and 3.2, Bayes' rule gives

$$r_k(X) = \sum_{\ell=1}^{K} (A^g)_{k\ell} \, \tilde{\pi}_\ell(X), \qquad A^g := ((T^g)^\top)^{-1}.$$

Hence

$$f(X)_k - r_k(X) = f(X)_k - \tilde{\pi}_k(X) + \underbrace{\sum_{\ell=1}^{K} (\mathbf{1}_{k=\ell} - A^g_{k\ell}) \, \tilde{\pi}_\ell(X)}_{=:s^g_k(X)}.$$

By definition of $s^g_k(X)$ we have the decomposition

$$f(X)_k - r_k(X) = a_k(X) + s^g_k(X), \qquad a_k(X) := f(X)_k - \tilde{\pi}_k(X).$$

Next, introduce the $L^p$-norm over the product space $(X, k)$ (conditionally on $G = g$):

$$\|v\|_p := \Big( \frac{1}{K} \sum_{k=1}^{K} \mathbb{E}[\,|v_k(X)|^p\,] \Big)^{1/p}.$$

With this notation

$$\|a\|_p = \widetilde{\mathrm{CE}}_g(f), \quad \|s^g\|_p = \Big( \frac{1}{K} \sum_{k=1}^{K} \mathbb{E}[\,|s^g_k(X)|^p\,] \Big)^{1/p}, \quad \|a + s^g\|_p = \mathrm{CE}_g(f).$$

The $L^p$ space ($p \geq 1$) satisfies the Minkowski inequality:

$$\|a + s^g\|_p \geq \big| \|a\|_p - \|s^g\|_p \big|.$$

Substituting the three norms identified above yields

$$\mathrm{CE}_g(f) \geq \big| \widetilde{\mathrm{CE}}_g(f) - \|s^g\|_p \big| = \Big| \widetilde{\mathrm{CE}}_g(f) - \Big( \frac{1}{K} \sum_{k=1}^{K} \mathbb{E}[\,|s^g_k(X)|^p\,] \Big)^{1/p} \Big|,$$

which is exactly the claimed lower bound. $\qquad\qquad\qquad\qquad\qquad\qquad\qquad\qquad\qquad\square$

### A.4   Proof of Theorem 4.1

**Closed-Form Decomposition of the Sufficiency Gap.**   Let $s = f(X) \in \Delta^{K-1}$ be the model score, $\tilde{\pi}_g(s) = \mathbb{P}(\tilde{Y} = \cdot \mid f(X) = s, G = g)$ the group-conditional noisy label posterior, and $\tilde{\pi}(s) = \mathbb{P}(\tilde{Y} = \cdot \mid f(X) = s)$ the marginal counterpart. Let $T \in [0, 1]^{K \times K}$ where $T_{i,j} = P(\tilde{Y} = j \mid Y = i) = \sum_{g \in \mathcal{G}} T^g_{i,j} P(G = g)$. Then under Assumptions 3.1 and 3.2, for any $p \geq 1$,

$$\mathrm{Suf}_g(f) = \left( \frac{1}{K} \mathbb{E} \left[ \left\| \underbrace{((T^g)^\top)^{-1}(\tilde{\pi}_g(f(X)) - \tilde{\pi}(f(X)))}_{\text{posterior shift}} + \underbrace{(((T^g)^\top)^{-1} - (T^\top)^{-1}) \, \tilde{\pi}(f(X))}_{\text{noise matrix mismatch}} \right\|^p_p \right] \right)^{1/p}.$$

*Proof.* Let $s = f(X) \in \Delta^{K-1}$ and set

$$\pi_g(s) = P(Y = \cdot \mid s, G = g), \quad \pi(s) = P(Y = \cdot \mid s),$$
$$\tilde{\pi}_g(s) = P(\tilde{Y} = \cdot \mid s, G = g), \quad \tilde{\pi}(s) = P(\tilde{Y} = \cdot \mid s).$$

From Definition 3.1, we have

$$\mathrm{Suf}_g(f)^p = \frac{1}{K}\mathbb{E}[\|\pi_g(s) - \pi(s)\|_p^p \mid G = g]. \tag{9}$$

Assumption 3.1 implies

$$\tilde{\pi}_g(s) = (T^g)^\top \pi_g(s) \quad \text{and} \quad \tilde{\pi}(s) = T^\top \pi(s) \tag{10}$$

and Assumption 3.2 gives

$$\pi_g(s) = ((T^g)^\top)^{-1}\tilde{\pi}_g(s) \quad \text{and} \quad \pi(s) = (T^\top)^{-1}\tilde{\pi}(s). \tag{11}$$

Subtracting the two identities in (11),

$$\pi_g(s) - \pi(s) = ((T^g)^\top)^{-1}(\tilde{\pi}_g(s) - \tilde{\pi}(s)) + (((T^g)^\top)^{-1} - (T^\top)^{-1})\tilde{\pi}(s). \tag{12}$$

Substituting (12) into (9) yields

$$\mathrm{Suf}_g(f) = \left(\frac{1}{K}\mathbb{E}[\|((T^g)^\top)^{-1}(\tilde{\pi}_g(s) - \tilde{\pi}(s)) + (((T^g)^\top)^{-1} - (T^\top)^{-1})\tilde{\pi}(s)\|_p^p \mid G = g]\right)^{1/p}. \tag{13}$$

Equation (13) is exactly the closed-form decomposition stated in Theorem 4.1. □

## A.5 Errors in Estimation of $T^g$

Finally, estimating $T$ from finite samples can be difficult. Proposition A.4 bounds the true sufficiency gap error with respect to the empirical sufficiency gap, mediated by estimating $T, T^g$ from finite samples.

**Lemma A.3** (Concentration of transition matrix estimates)**.** Let $\hat{T}^g$ and $\hat{T}$ be the empirical frequency-count estimates of the true transition matrices $T^g$ and $T$ from $n_g$ and $n$ i.i.d. samples, respectively. Assume each class appears at least $n_g/K$ (resp. $n/K$) times. Then for any $\delta \in (0,1)$, with probability at least $1 - \delta$,

$$\|T^g - \hat{T}^g\|_2 \le \eta_g, \qquad \|T - \hat{T}\|_2 \le \eta,$$

where

$$\eta_g = \sqrt{\frac{2K^2 \log(2K/\delta)}{n_g}}, \qquad \eta = \sqrt{\frac{2K^2 \log(2K/\delta)}{n}}.$$

*Proof.* Fix $i \in [K]$ and let $m_i \ge n_g/K$ be the number of samples with $Y = i$. By a vector-valued Hoeffding bound,

$$\mathbb{P}(\|\hat{T}_{i,\cdot}^g - T_{i,\cdot}^g\|_2 \ge \varepsilon) \le 2\exp\left(-\frac{m_i \varepsilon^2}{2}\right) \le 2\exp\left(-\frac{n_g \varepsilon^2}{2K}\right).$$

Taking a union bound over $i \in [K]$ gives

$$\mathbb{P}\left(\max_i \|\hat{T}_{i,\cdot}^g - T_{i,\cdot}^g\|_2 \ge \varepsilon\right) \le 2K \exp\left(-\frac{n_g \varepsilon^2}{2K}\right).$$

Setting $\varepsilon = \sqrt{\frac{2K \log(2K/\delta)}{n_g}}$ ensures the RHS is at most $\delta$. Then using $\|T^g - \hat{T}^g\|_2 \le \|T^g - \hat{T}^g\|_F \le \sqrt{K}\max_i \|\hat{T}_{i,\cdot}^g - T_{i,\cdot}^g\|_2$, we have

$$\|T^g - \hat{T}^g\|_2 \le \sqrt{K} \cdot \sqrt{\frac{2K \log(2K/\delta)}{n_g}} = \sqrt{\frac{2K^2 \log(2K/\delta)}{n_g}}.$$

The same argument applies to $\|T - \hat{T}\|_2$, completing the proof. □

**Proposition A.4.** Suppose Assumptions 3.1 and 3.2 hold. Let $\hat{T}^g$ and $\hat{T}$ be estimates of $T^g$ and $T$, respectively, such that with probability at least $1 - \delta$,

$$\|T^g - \hat{T}^g\|_2 \leq \eta_g \quad \text{and} \quad \|T - \hat{T}\|_2 \leq \eta.$$

Assume the estimates are well-conditioned, i.e., $\alpha_g \eta_g < 1$ and $\alpha \eta < 1$, where $\alpha_g = \|((T^g)^\top)^{-1}\|_2$ and $\alpha = \|(T^\top)^{-1}\|_2$. Define

$$C_g = \frac{\alpha_g^2}{1 - \alpha_g \eta_g} \quad \text{and} \quad C = \frac{\alpha^2}{1 - \alpha \eta}.$$

Then for some constant $M_p$ depending only on $p$ (from the sufficiency gap definition), with probability at least $1 - \delta$,

$$\left| \mathrm{Suf}_g(f) - \widehat{\mathrm{Suf}}_g(f) \right| \leq \left( M_p C_g \eta_g + C\eta \right)_+, \quad \text{where} \quad (x)_+ = \max\{0, x\}.$$

*Proof.* Let $\mathcal{E} = \{\|T^g - \hat{T}^g\|_2 \leq \eta_g, \ \|T - \hat{T}\|_2 \leq \eta\}$. By Lemma A.3, $\mathbb{P}(\mathcal{E}) \geq 1 - \delta$, and we work on this event.

Let $A^g = ((T^g)^\top)^{-1}$ and $B^g = A^g - (T^\top)^{-1}$, and define their empirical analogues $\hat{A}^g = ((\hat{T}^g)^\top)^{-1}$ and $\hat{B}^g = \hat{A}^g - (\hat{T}^\top)^{-1}$. By the Neumann-series lemma, for any matrix $A$ and perturbation $\Delta$ such that $\|A^{-1}\|_2 \|\Delta\|_2 < 1$, we have

$$\|(A + \Delta)^{-1} - A^{-1}\|_2 \leq \frac{\|A^{-1}\|_2^2 \|\Delta\|_2}{1 - \|A^{-1}\|_2 \|\Delta\|_2}.$$

Applying this to $T^g$ and $T$, and using the definitions $\alpha_g = \|A^g\|_2$ and $\alpha = \|(T^\top)^{-1}\|_2$, we obtain

$$\|\hat{A}^g - A^g\|_2 \leq \frac{\alpha_g^2 \eta_g}{1 - \alpha_g \eta_g} = C_g \eta_g \quad \text{and} \quad \|(\hat{T}^\top)^{-1} - (T^\top)^{-1}\|_2 \leq \frac{\alpha^2 \eta}{1 - \alpha \eta} = C\eta.$$

Now define the vector-valued sufficiency for group $g$ as

$$\mathbf{z}_g = A^g \Delta_{\tilde{\pi}} + \tilde{\pi} B^g,$$

and similarly, define the empirical estimate

$$\hat{\mathbf{z}}_g = \hat{A}^g \Delta_{\tilde{\pi}} + \tilde{\pi} \hat{B}^g.$$

By triangle inequality and sub-multiplicativity of norms,

$$\|\hat{\mathbf{z}}_g - \mathbf{z}_g\|_p \leq \|\hat{A}^g - A^g\|_2 \|\Delta_{\tilde{\pi}}\|_p + \|\tilde{\pi}\|_p \|\hat{B}^g - B^g\|_2.$$

Note that $\|\Delta_{\tilde{\pi}}\|_p \leq 2$ and $\|\tilde{\pi}\|_p \leq 1$ since both are probability distributions. Moreover, by triangle inequality on the matrices,

$$\|\hat{B}^g - B^g\|_2 = \|\hat{A}^g - A^g + (T^\top)^{-1} - (\hat{T}^\top)^{-1}\|_2 \leq \|\hat{A}^g - A^g\|_2 + \|(\hat{T}^\top)^{-1} - (T^\top)^{-1}\|_2.$$

Combining the bounds above, we obtain

$$\|\hat{\mathbf{z}}_g - \mathbf{z}_g\|_p \leq 2C_g \eta_g + (C_g \eta_g + C\eta) = (2+1)C_g \eta_g + C\eta = M_p C_g \eta_g + C\eta,$$

for $M_p = 3$ (or more generally any $M_p \geq 2$).

We now translate this into a bound on the sufficiency gap. By Minkowski's inequality,

$$|\mathrm{Suf}_g(f) - \widehat{\mathrm{Suf}}_g(f)| \leq \left( \frac{1}{K} \mathbb{E} \|\hat{\mathbf{z}}_g - \mathbf{z}_g\|_p^p \right)^{1/p} \leq (M_p C_g \eta_g + C\eta)_+.$$

This bound holds on event $\mathcal{E}$, which occurs with probability at least $1 - \delta$. This completes the proof. □

## A.6   Proof of Corollary 4.1.1

Here, we prove the contrapositive of the corollary: Under Assumptions 3.1 and 3.2, if $\tilde{Y} \perp\!\!\!\perp G \mid f(X)$ and $T^g = T$ for all $g \in \mathcal{G}$, then $\mathrm{Suf}_g(f) = 0$ for all $g \in \mathcal{G}$.

*Proof.* Applying Theorem 4.1, $T^g = T$ implies that $\Delta_{T^g} = 0$, and $\tilde{Y} \perp\!\!\!\perp G \mid f(X)$ implies $\tilde{\pi}_g(s) = \tilde{\pi}(s)$ for all $s$ and $g$, and so $\Delta_{\tilde{\pi}}(f(X)) = 0$. Thus, $\mathrm{Suf}_g(f) = 0$. □

## A.7 Proof of Proposition 5.1

*Proof.* For clarity we give the proof for $p = 1$; the extension to $p > 1$ follows by replacing absolute values with $|\cdot|^p$ and using the generalized Pinsker inequality [95].

Fix a group $g \in \mathcal{G}$ and let $Z := f(X) \in \Delta^{K-1}$ be the (predicted) score vector. For every score value $z$, define the conditional label distributions

$$P_{Y|Z=z} \quad \text{and} \quad P_{Y|Z=z,G=g}.$$

Because $\sum_{k=1}^{K} \mathbf{1}_{Y=k} = 1$, the $L_1$-form of the per-group sufficiency gap (Definition 3.2 with $p = 1$) can be written as

$$\text{Suf}_g(f) \;=\; \frac{1}{K}\, \mathbb{E}_{Z|G=g}\big[\| P_{Y|Z} - P_{Y|Z,G=g}\|_1\big] \;=\; \frac{2}{K}\, \mathbb{E}_{Z|G=g}\big[\text{TV}(P_{Y|Z}, P_{Y|Z,G=g})\big],$$

where $\text{TV}(P,Q) = \frac{1}{2}\|P - Q\|_1$ denotes total variation.

For any two discrete distributions, $\text{TV}(P,Q) \le \sqrt{\frac{1}{2} D_{\text{KL}}(P\|Q)}$. Applying this point-wise and Jensen's inequality,

$$\text{Suf}_g(f) \;\le\; \frac{\sqrt{2}}{K}\, \sqrt{\mathbb{E}_{Z|G=g}\big[D_{\text{KL}}\big(P_{Y|Z,G=g} \,\|\, P_{Y|Z}\big)\big]}. \tag{A}$$

By the chain rule for KL divergences,

$$I(Y;G \mid Z) \;=\; \mathbb{E}_Z\Big[\sum_{g'} P(G = g' \mid Z)\, D_{\text{KL}}\big(P_{Y|Z,G=g'} \,\|\, P_{Y|Z}\big)\Big]. \tag{B}$$

Because $P(G = g \mid Z) \le 1$ for every $Z$,

$$I(Y;G \mid Z) \;\ge\; P(G = g)\, \mathbb{E}_{Z|G=g}\big[D_{\text{KL}}\big(P_{Y|Z,G=g} \,\|\, P_{Y|Z}\big)\big].$$

Plugging this equation into (A) gives

$$\text{Suf}_g(f) \;\le\; \frac{\sqrt{2}}{K}\, \sqrt{\frac{I(Y;G \mid Z)}{P(G = g)}},$$

which is the first inequality.

Let $p_g := P(G = g)$, and $a_g := \mathbb{E}_{Z|G=g}\big[D_{\text{KL}}\big(P_{Y|Z,G=g} \,\|\, P_{Y|Z}\big)\big]$. We have, for each group,

$$\text{Suf}_g(f) \;\le\; \frac{\sqrt{2}}{K}\, \sqrt{a_g}\,.$$

Hence

$$\text{Suf}(f) = \sum_g p_g\, \text{Suf}_g(f) \;\le\; \frac{\sqrt{2}}{K}\, \sum_g p_g \sqrt{a_g}. \tag{S1}$$

Write $u_g := \sqrt{p_g}$ and $v_g := \sqrt{p_g\, a_g}$. Then $u_g v_g = p_g \sqrt{a_g}$ and

$$\sum_g p_g \sqrt{a_g} \;=\; \sum_g u_g v_g.$$

By the Cauchy–Schwarz inequality,

$$\Big(\sum_g u_g v_g\Big)^2 \;\le\; \Big(\sum_g u_g^2\Big)\Big(\sum_g v_g^2\Big) \;=\; \Big(\sum_g p_g\Big)\Big(\sum_g p_g\, a_g\Big) \;=\; I(Y;G \mid Z).$$

Taking square roots yields

$$\sum_g p_g \sqrt{a_g} \;\le\; \sqrt{I(Y;G \mid Z)}. \tag{S2}$$

Substituting (S2) into (S1) gives

$$\text{Suf}(f) \;\le\; \frac{\sqrt{2}}{K}\, \sqrt{I\big(Y;G \mid f(X)\big)}\,.$$

$\square$

## A.8 Unbiasedness and Convergence of the CMI Estimator

**Lemma A.5** (Unbiasedness of $R(f_\theta)$)**.** Under the assumption that $\forall x_i \in \mathcal{B} : h_\phi(f_\theta(x_i)) = \mathbb{P}(Y \mid f_\theta(x_i))$ and $h_\psi(f_\theta(x_i)) = \mathbb{P}(Y \mid f_\theta(x_i), g_i)$, we have that

$$\mathbb{E}[R(f_\theta)] = I(Y; G \mid f(X))$$

*Proof.* Let $\mathbf{h}_Y(\cdot \mid s_i) := [-\log p_Y(1 \mid s_i), ..., -\log p_Y(K \mid s_i)]$, and $\mathbf{h}_{\tilde{Y}}(\cdot \mid s_i)$ similarly. Let $[\cdot]_{\tilde{y}_i}$ denote taking the $\tilde{y}_i$-th element of this $K$-dimensional vector.

To clarify the problem, we start with the definition of entropy:

$$H(Y \mid f(X)) = \mathbb{E}_{X,Y}[-\log p_Y(Y \mid f(X))],$$

which we can estimate given a minibatch $\{(x_i, g_i, \tilde{y}_i)_{i=1}^B\}$ by applying the (unbiased) Monte Carlo estimator, defining $s_i = f(x_i)$ for convenience:

$$\hat{H}(Y \mid f(X)) = \frac{1}{B} \sum_{i=1}^B -\log p_Y(y_i \mid s_i) = \frac{1}{B} \sum_{i=1}^B [\mathbf{h}_Y(\cdot \mid s_i)]_{y_i}.$$

Since we do not observe $y_i$, we propose instead to use the backward-correction estimator shown in Equation (4):

$$\hat{H}'(Y \mid f(X)) = \frac{1}{B} \sum_{i=1}^B [T_{g_i}^{-1} \mathbf{h}_Y(\cdot \mid s_i)]_{\tilde{y}_i}.$$

We want to show that this estimator is unbiased, i.e. $\mathbb{E}_{X,G,Y,\tilde{Y}}[\hat{H}'(Y \mid f(X))] = H(Y \mid f(X))$.

First, observe that

$$
\begin{aligned}
\mathbb{E}_{\tilde{Y}|Y=y_i, G=g_i}\big[\big(T_{g_i}^{-1}\mathbf{h}_Y(\cdot \mid s_i)\big)_{\tilde{Y}}\big] &= \sum_{c=1}^K \mathbb{P}(\tilde{Y}=c \mid Y=y_i, G=g_i) \left[T_{g_i}^{-1}\mathbf{h}_Y(\cdot \mid s_i)\right]_c \\
&= \sum_{c=1}^K T_{y_i,c}^{g_i} \left[T_{g_i}^{-1}\mathbf{h}_Y(\cdot \mid s_i)\right]_c \\
&= e_{y_i}^\top T_{g_i} T_{g_i}^{-1} \mathbf{h}_Y(\cdot \mid s_i) \\
&= \big[\mathbf{h}_Y(\cdot \mid s_i)\big]_{y_i} \\
&= -\log p_Y(y_i \mid s_i),
\end{aligned}
\tag{1}
$$

which is no longer a function of $g_i$. Similarly,

$$\mathbb{E}_{\tilde{Y}|Y,G}\big[T_G^{-1} \mathbf{h}_Y(\cdot \mid s_i)\big] = -\log p_Y(Y \mid s_i).$$

Then,

$$
\begin{aligned}
\mathbb{E}_{X,G,Y,\tilde{Y}}[\hat{H}'(Y \mid f(X))] &= \mathbb{E}_{X,G,Y}\big[\mathbb{E}_{\tilde{Y}|X,G,Y}[\hat{H}'(Y \mid f(X))]\big] \\
&= \mathbb{E}_{X,G,Y}\big[\mathbb{E}_{\tilde{Y}|G,Y}[\hat{H}'(Y \mid f(X))]\big] && \text{(Assumption 3.1)} \\
&= \mathbb{E}_{X,G,Y}[-\log p_Y(Y \mid f(X))] && \text{(Substituting (1))} \\
&= \mathbb{E}_{X,Y}[-\log p_Y(Y \mid f(X))] \\
&= H(Y \mid f(X)).
\end{aligned}
$$

Thus, $\hat{H}'$ is an unbiased estimator of $H(Y \mid f(X))$.

$\square$

**Lemma A.6** (Finite-sample convergence of the CMI estimator). Let $\hat{I}_n$ be the CMI estimator defined in Equation 8, estimated with IID samples $(x_i, g_i, \tilde{y}_i)_{i=1}^n$. Assume that $\forall x_i \in \mathcal{B} : h_\phi(f_\theta(x_i)) = \mathbb{P}(Y \mid f_\theta(x_i))$ and $h_\psi(f_\theta(x_i)) = \mathbb{P}(Y \mid f_\theta(x_i), g_i)$. Further assume that there exists $\epsilon \in (0, 1/2)$ such that for all $(s, y, g)$,

$$p_{Y|S}(y \mid s) \geq \epsilon, \quad p_{G|S}(g \mid s) \geq \epsilon, \quad p_{Y,G|S}(y, g \mid s) \geq \epsilon.$$

Then, under Assumptions 3.1 and 3.2, we have that, with probability $1 - \delta$,

$$|\hat{I}_n - I(Y; G \mid f(X))| \leq 3\sqrt{2} \log(1/\epsilon) \sqrt{\frac{\log(6/\delta)}{n}} = \mathcal{O}(n^{-1/2}).$$

Further, combining with Proposition 5.1, we have that

$$\mathrm{Suf}_g(f) \leq \sqrt{\frac{2}{K}} (\hat{I}_n + \mathcal{O}(n^{-1/2}))^{1/2}.$$

*Proof.* Let $S = f(X)$ and suppose the conditional probability heads are *oracle*, i.e., $\hat{p}_{Y|S} = p_{Y|S}$, $\hat{p}_{G|S} = p_{G|S}$, and $\hat{p}_{Y,G|S} = p_{Y,G|S}$ (approximation error $= 0$). Define the bounded random variables

$$\phi_Y := -\log p(Y \mid S), \qquad \phi_G := -\log p(G \mid S), \qquad \phi_{YG} := -\log p(Y, G \mid S).$$

By the $\varepsilon$-lower bound, each $\phi_\cdot \in [0, \log(1/\varepsilon)]$ almost surely. With $\mathbb{E}$ the population expectation and $\mathbb{E}_n$ the empirical mean over an i.i.d. sample of size $n$, the plug-in estimator and the population CMI can be written as

$$\hat{I}_n = \mathbb{E}_n[\phi_Y] + \mathbb{E}_n[\phi_G] - \mathbb{E}_n[\phi_{YG}], \qquad I(Y; G \mid S) = \mathbb{E}[\phi_Y] + \mathbb{E}[\phi_G] - \mathbb{E}[\phi_{YG}],$$

because $H(Y \mid S) = \mathbb{E}[\phi_Y]$, $H(G \mid S) = \mathbb{E}[\phi_G]$ and $H(Y, G \mid S) = \mathbb{E}[\phi_{YG}]$.

Therefore

$$\left|\hat{I}_n - I(Y; G \mid S)\right| \leq \left|(\mathbb{E}_n - \mathbb{E})[\phi_Y]\right| + \left|(\mathbb{E}_n - \mathbb{E})[\phi_G]\right| + \left|(\mathbb{E}_n - \mathbb{E})[\phi_{YG}]\right|.$$

Each term is a deviation of the empirical mean of a bounded random variable. By Hoeffding's inequality, for any $t > 0$ and any bounded $Z \in [0, B]$,

$$\Pr\left(\left|(\mathbb{E}_n - \mathbb{E})[Z]\right| \geq t\right) \leq 2 \exp\left(-\frac{2nt^2}{B^2}\right).$$

Applying this with $B = \log(1/\varepsilon)$ and a union bound over the three variables $\{\phi_Y, \phi_G, \phi_{YG}\}$, we obtain that with probability at least $1 - \delta$,

$$\left|\hat{I}_n - I(Y; G \mid S)\right| \leq 3 \log\frac{1}{\varepsilon} \sqrt{\frac{1}{2n} \log\frac{6}{\delta}}.$$

$\square$

# B   Experimental Details

## B.1   Data Processing

`income`   We use the ACS income dataset from Ding et al. [84], using data from all 50 states and Puerto Rico for 2018. Gender is used as the protected attribute, with women being the group with noise added. We create a 3-class classification problem from the dataset by defining class 1 to be less than \$30,000, class 2 to be between \$30,000 and \$50,000, and class 3 to be greater than \$50,000. This dataset is governed by the terms of use provided by the US Census Bureau.

`adult`   We use the Adult dataset from the UCI Machine Learning repository [80]. Gender is used as the protected attribute, with women being the group with noise added. This dataset is licensed under a Creative Commons Attribution 4.0 International (CC BY 4.0) license.

`crime`   We use the Communities and Crime dataset from the UCI Machine Learning repository [80]. Following Denis et al. [96], we create a 5-class classification problem by dividing the continuous crime rate into 5 equally sized bins. Following Denis et al. [96], we define the binary attribute to be whether a community is majority Black, with the non-Black group being noised. This dataset is licensed under a Creative Commons Attribution 4.0 International (CC BY 4.0) license.

`lsac`   We use the law school admissions dataset from Wightman [81], where the goal is to predict whether someone passes the bar given their academic history. We use race as the protected attribute, and add noise to White applicants.

`grades`   We use the dataset from Lenders and Calders [85], where the goal is to predict whether a high school student will pass an exam. We use the biased human labels from the ranking strategy as observed labels, and the eight categorical features from the Kaggle version of the dataset as features.

`cifar10ns`   We combine the CIFAR-10N dataset from Wei et al. [88], with the CIFAR-10S dataset from Wang et al. [89]. Specifically, we use the human-derived noisy labels from CIFAR-10N, and augment the images to create an attribute by randomly turning 5% of images belonging to the second class to be grayscale, and the remaining classes to be 95% grayscale. The attribute is then whether the image is grayscale or not. Note that CIFAR-10N is released under the Creative Commons Attribution-NonCommercial 4.0 license.

`civilcomments` **[86]**   We use the CivilComments dataset from the WILDS library [97], where the goal is to classify toxicity given a string of text. We choose toxicity as the binary target, and the presence of any identity as the protected attribute. Note that each sample in CivilComments has been labeled by multiple annotators. For the noisy labels, we observe labels with $Ber(p)$, where $p$ is the proportion of labelers who said that the sample is *severely toxic*. Note that severe toxicity is a different label than the toxicity true label.

`clothing1m` **[87]**   The objective to predict the type of clothing from an image scraped from the web. The noisy label is derived from keyword scraping, while the true label is derived from human labeling. Note that only the validation and test sets contain clean labels. To create an attribute for this dataset, we run an MTCNN [98] from the `facenet-pytorch` repo to detect the presence of a face in the image. We found that 64.8% of images in the dataset contain a face. The dataset is released for academic use under an unknown license.

## B.2   Synthetic Noise

For both synthetic noise types, we add noise to the labels for only one of the groups.

### B.2.1   Group Uniform

Given noise rate $\eta \in (0, 1)$, We noise one group with the uniform noise transition matrix:

$$
T = \begin{pmatrix}
1-\eta & \frac{\eta}{K-1} & \cdots & \frac{\eta}{K-1} \\
\frac{\eta}{K-1} & 1-\eta & \cdots & \frac{\eta}{K-1} \\
\vdots & \vdots & \ddots & \vdots \\
\frac{\eta}{K-1} & \frac{\eta}{K-1} & \cdots & 1-\eta
\end{pmatrix}
$$

### B.2.2   Group Asymmetric

Given noise rate $\eta \in (0, 1)$, for $K = 2$, we noise one group with the following transition matrix:

$$
T = \begin{pmatrix}
1 & 0 \\
\eta & 1-\eta
\end{pmatrix}
$$

For $K > 2$, we use the pairflip noise type from Zhu et al. [99]:

$$P = \begin{pmatrix} 1-\eta & \eta & 0 & \cdots & 0 \\ 0 & 1-\eta & \eta & \ddots & \vdots \\ \vdots & \ddots & \ddots & \ddots & 0 \\ 0 & \cdots & 0 & 1-\eta & \eta \\ \eta & 0 & \cdots & 0 & 1-\eta \end{pmatrix}$$

For both noise types, we use $\eta = 0.75$ for all datasets, except for `lsac` where we use $\eta = 0.4$. We choose these noise rates as they produce ERM models with better-than-chance performance, while having significant sufficiency gaps.

### B.3 Hyperparameter Grids

For all datasets and methods, we use a random hyperparameter search with 20 iterations.

We use the following hyperparameter grids which are common across all methods:

- Batch size: $2^{Rand(6,11)}$ for tabular datasets, $2^{Rand(4,6)}$ for text and image datasets
- Number of steps: $Rand(750, 2000)$ for tabular datasets, $Rand(2000, 4000)$ for text and image datasets.
- Learning rate: $10^{Unif(-5,-2)}$

For the tabular datasets, we use an MLP varying the following architecture hyperparameters:

- Depth: Rand(2, 5)
- Width: Rand(32, 256)
- Dropout: Unif(0, 0.5)

We use the following hyperparameter grids for specific base losses:

- GCE $q$: Unif (0, 1)
- ELR $\lambda$: Unif(1.0, 5.0)
- ELR $\beta$: Unif(0.5, 0.9)
- Peer loss $\alpha$: Unif(0, 2)

We use the following hyperparameter grids for the MC baseline:

- $\alpha$: $10^{Unif(-3,0)}$
- $\lambda$: $10^{Unif(-3,0)}$
- Number of iterations: Rand(50, 200)

We use the following hyperparameter grids for SUF-REG and CMI-REG:

- Learning rate for $h_\phi$ and $h_\psi$: $10^{Unif(-3,-2)}$
- $C$: Rand(1, 8)
- $\lambda$: $10^{Unif(-0.5,1)}$
- Warm up steps: Rand(20, 100)

## C    SUF-REG Method

In order to directly minimize Theorem 4.1, we parameterize $\tilde{\pi}(s)$ by a model $h_\phi(f_\theta(x_i))$, and $\tilde{\pi}_g(s)$ by a model $h_\psi(f_\theta(x_i), g)$, with both being linear layers operating on the logits. We then alternate learning $\psi$ and $\phi$ with learning $\theta$, similar to CMI-REG. The algorithm is found in Algorithm 2.

**Algorithm 2:** Sufficiency–Regularised Learning under Class–Attribute Label Noise

**Input:** Batch $\mathcal{B} = \{(\mathbf{x}_i, \mathbf{g}_i, \tilde{\mathbf{y}}_i)\}_{i=1}^{B}$ ;
Per–group transition matrices $\{T^g \in [0,1]^{K \times K}\}_{g \in \mathcal{G}}, \lambda > 0, C \geq 1$;
Learning rates $\alpha_f, \alpha_h > 0$;
Current classifier and auxiliary heads: $f_\theta, h_\phi, h_\psi$ ;
$\ell_{pred}(z_i, \tilde{y}_i)$;
**Output:** Updated $f_\theta, h_\phi, h_\psi$

1   $z_i \leftarrow f_\theta(x_i) \ \forall i \in \mathcal{B}$ ;        // logits
2   **for** $j = 1$ **to** $C$ **do**        // update heads, freeze $\theta$

3     $\phi \leftarrow \phi - \alpha_h \, \nabla_\phi \dfrac{1}{B} \sum_{i \in \mathcal{B}} \big(\log h_\phi(z_i)\big)_{\tilde{y}_i}$ ;

4     $\psi \leftarrow \psi - \alpha_h \, \nabla_\psi \dfrac{1}{B} \sum_{i \in \mathcal{B}} \big(\log h_\psi(z_i, g_i)\big)_{\tilde{y}_i}$ ;

5   $\mathcal{L}_{\text{pred}} = \dfrac{1}{B} \sum_{i \in \mathcal{B}} \ell_{\text{pred}}(z_i, \tilde{y}_i)$;

6   $R_B = \dfrac{1}{K} \sum_{i \in \mathcal{B}} \big| ((T^{g_i})^\top)^{-1} h_\psi(z_i, g_i) - (T^\top)^{-1} h_\phi(z_i) \big|$;

7   $\mathcal{L} = \mathcal{L}_{\text{pred}} + \lambda R_B$
8   $\theta \leftarrow \theta - \alpha_f \, \nabla_\theta \mathcal{L}$;
9   **return** $f_\theta, h_\phi, h_\psi$;

# D   Additional Experimental Results

## D.1   Varying Synthetic Noise Type

In Tables 2 and 3, we present results for group-asymmetric and group-uniform synthetic noise respectively at the default noise levels. Across eight combinations of dataset and noise types, we find that our regularizer achieves the lowest sufficiency gap in all cases (with CMI-REG being the best in seven of eight cases).

Table 3: Clean test-set AUROC and sufficiency gap (Suf) of our regularizers when added to off-the-shelf loss functions, for tabular datasets with synthetic **group-uniform** noise and known transition matrices. Models are selected using the clean Brier score. The best method within each base loss is **bolded**, and the best model overall is also **underlined**.

| | income | | adult | | lsac | | crime | |
|---|---|---|---|---|---|---|---|---|
| | AUROC ↑ | Suf ↓ | AUROC ↑ | Suf ↓ | AUROC ↑ | Suf ↓ | AUROC ↑ | Suf ↓ |
| CE | 0.762 (0.002) | 0.046 (0.000) | 0.634 (0.014) | 0.112 (0.003) | 0.587 (0.049) | 0.108 (0.009) | 0.613 (0.019) | 0.089 (0.001) |
| CE + SUF-REG | 0.670 (0.000) | **0.025** (0.003) | **0.755** (0.050) | **0.036** (0.010) | 0.563 (0.008) | 0.114 (0.006) | **0.641** (0.013) | 0.096 (0.006) |
| CE + CMI-REG | **0.795** (0.007) | 0.058 (0.004) | 0.698 (0.137) | 0.073 (0.018) | **0.596** (0.005) | **0.101** (0.008) | 0.625 (0.020) | **0.085** (0.010) |
| Forward | **0.761** (0.003) | 0.043 (0.001) | 0.539 (0.082) | 0.100 (0.013) | 0.457 (0.093) | **0.095** (0.004) | 0.602 (0.040) | **0.081** (0.018) |
| Forward + SUF-REG | 0.677 (0.001) | **0.024** (0.003) | **0.848** (0.006) | 0.039 (0.015) | **0.500** (0.000) | 0.107 (0.007) | **0.639** (0.004) | 0.086 (0.002) |
| Forward + CMI-REG | 0.744 (0.021) | 0.031 (0.005) | 0.846 (0.009) | **0.029** (0.010) | 0.496 (0.006) | 0.107 (0.007) | 0.618 (0.010) | 0.093 (0.004) |
| Backward | 0.764 (0.000) | 0.042 (0.002) | 0.540 (0.081) | 0.100 (0.013) | 0.476 (0.106) | 0.096 (0.011) | 0.605 (0.040) | 0.079 (0.008) |
| Backward + SUF-REG | 0.766 (0.001) | 0.048 (0.000) | 0.696 (0.013) | 0.045 (0.002) | 0.459 (0.097) | **0.092** (0.010) | 0.609 (0.003) | **0.074** (0.012) |
| Backward + CMI-REG | **0.768** (0.005) | **0.035** (0.001) | **0.878** (0.023) | **0.028** (0.008) | **0.523** (0.113) | 0.093 (0.013) | **0.630** (0.049) | 0.082 (0.010) |
| DMI | 0.617 (0.032) | 0.039 (0.013) | 0.386 (0.114) | **0.098** (0.010) | 0.474 (0.183) | **0.088** (0.002) | 0.419 (0.007) | 0.102 (0.003) |
| DMI + SUF-REG | 0.617 (0.032) | **0.038** (0.014) | 0.707 (0.040) | 0.106 (0.003) | 0.477 (0.175) | **0.088** (0.000) | 0.444 (0.038) | **0.097** (0.004) |
| DMI + CMI-REG | **0.619** (0.033) | 0.039 (0.010) | **0.757** (0.042) | 0.099 (0.008) | **0.478** (0.178) | **0.088** (0.000) | **0.477** (0.057) | 0.102 (0.017) |
| ELR | 0.767 (0.001) | 0.048 (0.002) | 0.555 (0.110) | 0.099 (0.017) | **0.363** (0.265) | 0.061 (0.051) | 0.610 (0.035) | **0.090** (0.005) |
| ELR + SUF-REG | 0.764 (0.003) | 0.045 (0.002) | 0.685 (0.157) | 0.079 (0.027) | 0.340 (0.007) | 0.096 (0.004) | **0.616** (0.027) | 0.092 (0.016) |
| ELR + CMI-REG | **0.784** (0.021) | 0.031 (0.014) | **0.806** (0.067) | 0.060 (0.019) | 0.230 (0.026) | **0.041** (0.003) | 0.599 (0.042) | 0.096 (0.005) |
| GCE | **0.768** (0.002) | 0.047 (0.004) | 0.540 (0.085) | 0.100 (0.013) | 0.459 (0.043) | 0.098 (0.003) | 0.604 (0.039) | 0.085 (0.014) |
| GCE + SUF-REG | 0.722 (0.056) | 0.019 (0.003) | 0.701 (0.010) | 0.061 (0.011) | 0.582 (0.013) | 0.103 (0.005) | **0.637** (0.009) | 0.081 (0.015) |
| GCE + CMI-REG | 0.747 (0.007) | **0.011** (0.001) | **0.848** (0.044) | 0.043 (0.020) | **0.704** (0.000) | **0.086** (0.005) | 0.609 (0.015) | **0.072** (0.001) |
| Peer | **0.767** (0.000) | 0.045 (0.004) | 0.510 (0.003) | **0.092** (0.004) | 0.556 (0.066) | 0.110 (0.008) | 0.616 (0.001) | 0.077 (0.023) |
| Peer + SUF-REG | 0.745 (0.039) | 0.024 (0.011) | 0.509 (0.003) | **0.092** (0.004) | **0.640** (0.175) | **0.082** (0.025) | **0.670** (0.025) | **0.061** (0.008) |
| Peer + CMI-REG | 0.742 (0.013) | **0.019** (0.002) | **0.756** (0.021) | 0.098 (0.003) | 0.590 (0.052) | 0.114 (0.011) | 0.606 (0.031) | 0.102 (0.005) |
| MC | **0.805** (0.002) | 0.030 (0.002) | 0.861 (0.015) | 0.045 (0.011) | 0.500 (0.000) | 0.107 (0.007) | 0.508 (0.013) | 0.101 (0.008) |
| GroupPeer | – | – | 0.572 (0.062) | 0.076 (0.004) | 0.635 (0.056) | 0.100 (0.006) | – | – |
| Oracle | 0.854 (0.001) | 0.003 (0.000) | 0.919 (0.002) | 0.013 (0.001) | 0.838 (0.011) | 0.018 (0.009) | 0.825 (0.013) | 0.046 (0.010) |

## D.2 Real-World Noise

In Table 4, we present results for datasets with real noise. We find that in the `grades` dataset, CMI-REG does show significant improvements in sufficiency gap without much decrease in performance. However, in the other 3 datasets, the sufficiency gap is negligible when training with just cross-entropy on noisy data. In addition, almost none of the other loss functions outperform simple cross-entropy with significance on overall AUROC. This indicates that real-world noise is highly non-class-conditional, and existing learning-under-label-noise methods fail as a result. Further, it indicates that there the group attribute or noisy label have been selected when processing the datasets in a way such that there is actually *minimal label bias*. Regardless, in these scenarios, we would hope that applying our method does not have a negative effect, i.e. that it leaves AUROC unchanged while maintaining a negligbile sufficiency gap. We observe that this is the case for all three datasets.

Table 4: Clean test-set AUROC and sufficiency gap (Suf) for four datasets with real-world label noise. Models are selected using the clean Brier score. Within each base loss, the best value for every metric/dataset is **bolded**; the overall best non-Oracle value for each metric/dataset is additionally **underlined**.

| | grades | | civilcomments | | cifar10ns | | clothing1m | |
|---|---|---|---|---|---|---|---|---|
| | AUROC ↑ | Suf ↓ | AUROC ↑ | Suf ↓ | AUROC ↑ | Suf ↓ | AUROC ↑ | Suf ↓ |
| CE | 0.681 (0.043) | 0.050 (0.028) | **0.929** (0.001) | **0.004** (0.001) | 0.984 (0.001) | 0.009 (0.003) | **0.961** (0.001) | **0.007** (0.001) |
| CE + CMI-REG | **0.683** (0.049) | **0.047** (0.030) | **0.929** (0.001) | 0.005 (0.001) | 0.984 (0.000) | **0.007** (0.001) | 0.952 (0.002) | **0.007** (0.001) |
| Forward | **0.689** (0.048) | 0.055 (0.021) | **0.906** (0.008) | **0.010** (0.009) | **0.988** (0.000) | 0.003 (0.000) | 0.943 (0.001) | **0.009** (0.000) |
| Forward + CMI-REG | 0.674 (0.044) | **0.054** (0.014) | 0.904 (0.010) | 0.011 (0.008) | 0.987 (0.001) | **0.002** (0.000) | **0.944** (0.003) | **0.009** (0.002) |
| Backward | **0.692** (0.032) | 0.078 (0.002) | 0.883 (0.028) | 0.013 (0.011) | 0.976 (0.000) | **0.006** (0.000) | 0.931 (0.002) | **0.007** (0.000) |
| Backward + CMI-REG | 0.688 (0.058) | **0.051** (0.007) | **0.920** (0.002) | **0.004** (0.001) | 0.976 (0.001) | **0.006** (0.000) | **0.931** (0.002) | **0.007** (0.000) |
| ELR | **0.667** (0.035) | 0.060 (0.012) | **0.928** (0.002) | **0.004** (0.001) | 0.978 (0.004) | **0.003** (0.001) | **0.949** (0.000) | 0.008 (0.001) |
| ELR + CMI-REG | 0.666 (0.040) | **0.057** (0.009) | **0.928** (0.000) | **0.004** (0.000) | 0.976 (0.002) | **0.003** (0.001) | 0.942 (0.000) | **0.007** (0.000) |
| GCE | **0.683** (0.026) | 0.073 (0.017) | **0.915** (0.001) | **0.006** (0.000) | 0.986 (0.001) | 0.008 (0.002) | **0.942** (0.001) | **0.007** (0.000) |
| GCE + CMI-REG | 0.681 (0.053) | **0.048** (0.009) | 0.914 (0.002) | 0.009 (0.005) | **0.987** (0.000) | **0.004** (0.000) | 0.940 (0.001) | **0.007** (0.000) |
| Peer | 0.681 (0.035) | 0.067 (0.020) | **0.925** (0.004) | **0.005** (0.003) | 0.961 (0.002) | 0.004 (0.000) | **0.895** (0.000) | **0.009** (0.000) |
| Peer + CMI-REG | **0.692** (0.043) | **0.053** (0.018) | 0.924 (0.005) | 0.007 (0.004) | **0.962** (0.002) | **0.003** (0.000) | 0.894 (0.001) | **0.009** (0.000) |
| Oracle | 0.713 (0.075) | 0.050 (0.009) | 0.936 (0.001) | 0.004 (0.001) | – | – | – | – |

## D.3 Pareto Plots

To examine the trade-off between AUROC and sufficiency gap, we plot Pareto plots for these two metrics for each dataset with synthetic noise. We present two styles of Pareto plots. First, in Figure 3, we show each base loss in a subplot, in order to examine whether our methods consistently improve the baseline for each individual loss. We note that this is a fairly strict criteria, as there may be base losses that are incompatible with our loss. Regardless, we find that this occurs in most cases. Next, in Figure 4, we aggregate all losses in a single plot, in order to show the best trade-off in the state-of-the-art before our work (i.e. base losses only) versus the two regularizers we contribute. We find that our regularizers advance the state-of-the-art by having either one Pareto-dominating the base loss in all cases.

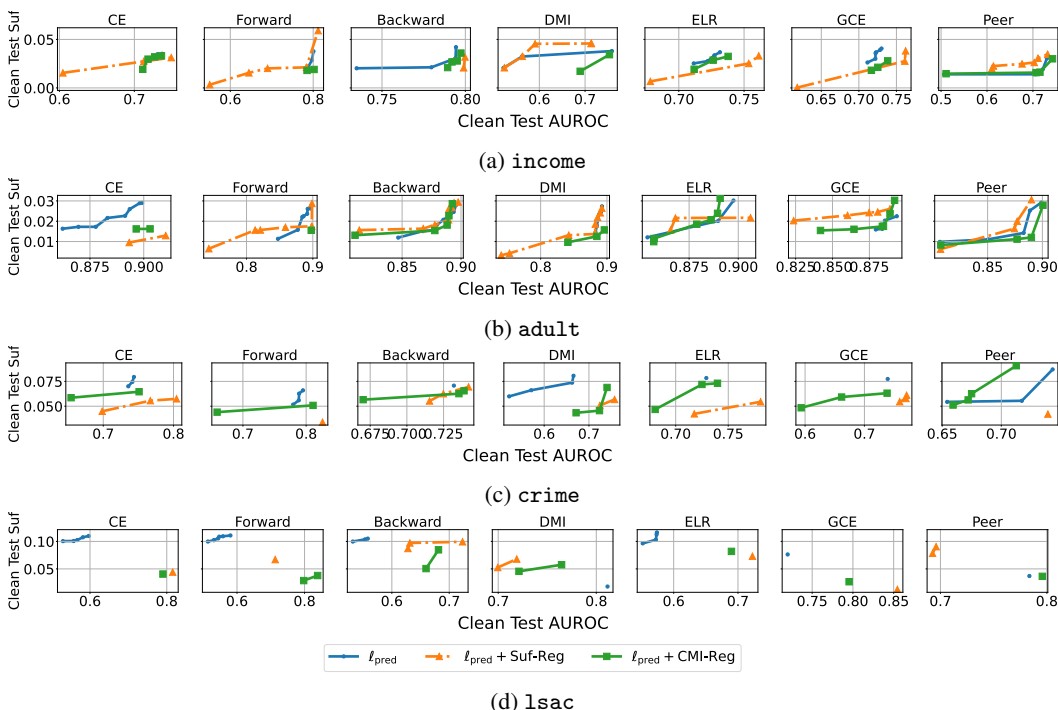

Figure 3: Pareto fronts of AUROC versus Suf of CMI-REG, SUF-REG, and baseline loss for synthetic group asymmetric noise and known transition matrices. A curve towards the bottom right of the plot is desirable as it achieves the best fairness-accuracy trade-off. We observe that CMI-REG and SUF-REG Pareto-dominate the base loss for most losses.

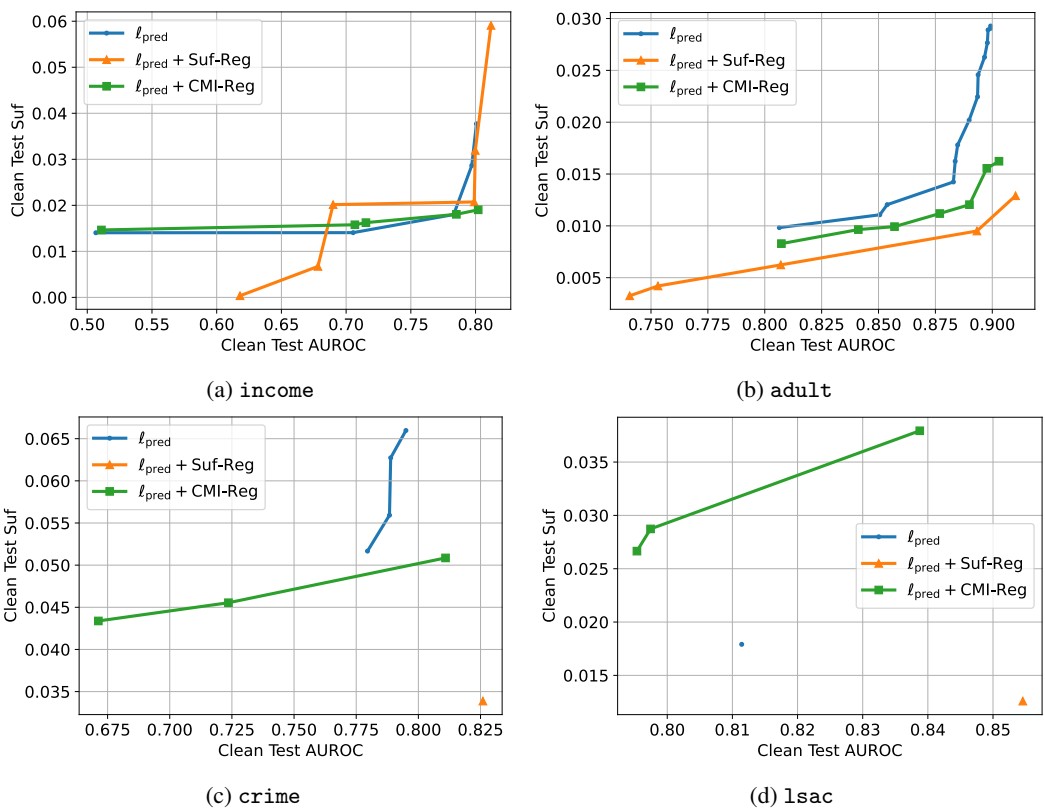

Figure 4: Pareto fronts of AUROC versus Suf of CMI-REG, SUF-REG, and baseline loss for synthetic group asymmetric noise and known transition matrices. A curve towards the bottom right of the plot is desirable as it achieves the best fairness-accuracy trade-off.

### D.4 Impact of Model Selection

In Table 5, we compare the performance of different model selection criteria. Selection criteria are: (1) *Clean Brier*: We select the model with the lowest overall Brier score on the *clean* validation set, which requires a validation set with clean labels. (2) *Default*: We use default hyperparameter values reported in the papers for hyperparameters in $\ell_{pred}$, and use $\lambda = 1, C = 3$ for SUF-REG and CMI-REG. (3) *Theory*: We select the model with the smallest value from Theorem 4.1 on the noisy validation set, which requires knowledge of the transition matrices but not any clean labels (4) *Noisy Brier*: We select the model with the lowest overall Brier score on the noisy validation set.

As expected, selecting using clean Brier gives the best performance and lowest sufficiency gap. Selecting using the theoretical expression (Theorem 4.1) on the noisy validation set gives similarly low sufficiency gaps, but compromises significantly on performance. Selecting using the noisy Brier gives mediocre performance with slightly worse sufficiency. Across all selection criteria, selected models with regularization terms have lower sufficiency gaps than the base model in almost all cases.

### D.5 Estimated Transition Matrix at Varying Noise Types

In Tables 6 and 7, we compare our method versus the base losses when estimated transition matrices are used at two other synthetic noise strengths for group-uniform noise ($\eta = 0.6$ and $\eta = 0.8$ respectively), with clean Brier as the selection criteria. We find that with the estimated transition matrix, regularizers still generally improve fairness, though the performance improvements become limited relative to when the real transition matrix is used.

In Tables 8 and 9, we show the performance and fairness of our method versus the base losses in the hardest setting: when using the estimated transition matrix (via Confident Learning [54]), and when using the noisy Brier score as the selection metric. We find that even in this challenging setting where

Table 5: Comparison of AUROC and Suf under synthetic Group Uniform (GU) and Group Asymmetric (GA) noise. Each entry is mean (std) across seven base losses.

| | | | AUROC | | | | Suf | | | |
|---|---|---|---|---|---|---|---|---|---|---|
| | | | Clean Brier | Default | Noisy Brier | Theory | Clean Brier | Default | Noisy Brier | Theory |
| income | GU | Base | **0.744** (0.054) | 0.642 (0.025) | 0.639 (0.023) | **0.645** (0.096) | 0.044 (0.005) | 0.047 (0.013) | 0.047 (0.009) | **0.029** (0.012) |
| | | Base + Suf-Reg | 0.709 (0.058) | 0.666 (0.019) | 0.672 (0.061) | 0.568 (0.092) | **0.032** (0.012) | 0.025 (0.006) | 0.030 (0.009) | 0.040 (0.013) |
| | | Base + CMI-Reg | 0.729 (0.060) | **0.704** (0.064) | **0.694** (0.036) | 0.597 (0.119) | 0.035 (0.011) | **0.024** (0.014) | **0.019** (0.008) | 0.036 (0.010) |
| | GA | Base | 0.730 (0.074) | 0.677 (0.057) | 0.681 (0.059) | **0.643** (0.101) | 0.040 (0.007) | 0.058 (0.020) | 0.058 (0.012) | **0.030** (0.009) |
| | | Base + Suf-Reg | **0.738** (0.076) | **0.689** (0.062) | **0.711** (0.045) | 0.558 (0.069) | 0.039 (0.012) | 0.059 (0.025) | **0.056** (0.013) | 0.037 (0.028) |
| | | Base + CMI-Reg | 0.731 (0.074) | 0.686 (0.061) | 0.678 (0.059) | 0.606 (0.107) | **0.035** (0.011) | **0.054** (0.023) | 0.058 (0.014) | 0.035 (0.015) |
| adult | GU | Base | 0.529 (0.093) | 0.619 (0.032) | 0.650 (0.048) | 0.640 (0.128) | 0.100 (0.010) | 0.109 (0.013) | 0.107 (0.008) | 0.097 (0.008) |
| | | Base + Suf-Reg | 0.700 (0.108) | **0.656** (0.103) | 0.530 (0.126) | 0.667 (0.162) | 0.065 (0.028) | **0.089** (0.026) | 0.095 (0.008) | 0.068 (0.028) |
| | | Base + CMI-Reg | **0.836** (0.063) | 0.640 (0.075) | **0.684** (0.073) | **0.706** (0.205) | **0.053** (0.032) | 0.095 (0.028) | **0.091** (0.012) | **0.067** (0.032) |
| | GA | Base | **0.893** (0.006) | 0.869 (0.032) | **0.893** (0.008) | **0.876** (0.030) | 0.026 (0.004) | 0.022 (0.004) | 0.026 (0.006) | 0.024 (0.007) |
| | | Base + Suf-Reg | 0.886 (0.009) | **0.876** (0.033) | 0.886 (0.009) | 0.802 (0.171) | 0.028 (0.004) | 0.025 (0.006) | 0.027 (0.005) | **0.019** (0.011) |
| | | Base + CMI-Reg | 0.892 (0.005) | 0.867 (0.033) | 0.891 (0.011) | 0.818 (0.181) | **0.023** (0.006) | **0.021** (0.005) | **0.025** (0.004) | 0.021 (0.006) |
| crime | GU | Base | 0.581 (0.072) | 0.608 (0.062) | 0.592 (0.053) | **0.585** (0.072) | 0.086 (0.012) | 0.107 (0.016) | **0.086** (0.011) | 0.087 (0.011) |
| | | Base + Suf-Reg | **0.608** (0.074) | **0.609** (0.049) | **0.594** (0.086) | 0.572 (0.092) | **0.084** (0.015) | 0.107 (0.018) | 0.090 (0.019) | **0.082** (0.029) |
| | | Base + CMI-Reg | 0.572 (0.054) | 0.589 (0.062) | 0.552 (0.091) | 0.552 (0.094) | 0.102 (0.016) | **0.105** (0.016) | 0.101 (0.009) | 0.097 (0.009) |
| | GA | Base | 0.676 (0.125) | 0.614 (0.084) | 0.673 (0.129) | 0.612 (0.126) | 0.077 (0.021) | 0.096 (0.021) | 0.081 (0.014) | **0.085** (0.027) |
| | | Base + Suf-Reg | **0.714** (0.140) | **0.636** (0.124) | **0.685** (0.102) | 0.618 (0.119) | **0.066** (0.017) | **0.082** (0.027) | 0.082 (0.017) | 0.088 (0.024) |
| | | Base + CMI-Reg | 0.687 (0.135) | 0.624 (0.102) | 0.677 (0.110) | **0.634** (0.134) | 0.070 (0.018) | 0.092 (0.023) | **0.074** (0.021) | 0.088 (0.021) |
| lsac | GU | Base | 0.482 (0.123) | **0.538** (0.038) | 0.487 (0.119) | 0.442 (0.113) | **0.094** (0.022) | 0.109 (0.007) | 0.095 (0.024) | 0.093 (0.009) |
| | | Base + Suf-Reg | 0.509 (0.120) | 0.518 (0.054) | 0.466 (0.099) | **0.539** (0.174) | 0.097 (0.014) | **0.100** (0.011) | 0.094 (0.022) | **0.082** (0.022) |
| | | Base + CMI-Reg | **0.559** (0.139) | 0.529 (0.043) | 0.425 (0.174) | 0.534 (0.167) | **0.094** (0.019) | 0.102 (0.008) | **0.079** (0.030) | 0.088 (0.013) |
| | GA | Base | 0.478 (0.116) | **0.542** (0.034) | 0.512 (0.088) | **0.533** (0.042) | 0.095 (0.021) | 0.110 (0.006) | 0.102 (0.010) | 0.111 (0.006) |
| | | Base + Suf-Reg | **0.594** (0.107) | 0.536 (0.070) | **0.514** (0.095) | 0.523 (0.054) | 0.095 (0.008) | 0.108 (0.007) | 0.101 (0.010) | 0.109 (0.012) |
| | | Base + CMI-Reg | 0.564 (0.165) | 0.538 (0.054) | 0.439 (0.164) | 0.442 (0.139) | **0.085** (0.028) | **0.107** (0.007) | **0.084** (0.033) | **0.086** (0.035) |

Table 6: Clean test-set AUROC and sufficiency gap (Suf) of our regularizers when added to off-the-shelf loss functions, for tabular datasets with synthetic **group–uniform** noise at $\eta = 0.6$ and real vs. **estimated** transition matrices. Models are selected using the *clean Brier score*. The best method within each base loss is **bolded**; the overall best non-Oracle model for each metric/dataset is additionally **underlined**.

| | Real $T_g$ | | | | Estimated $T_g$ | | | |
|---|---|---|---|---|---|---|---|---|
| | income | | adult | | income | | adult | |
| | AUROC ↑ | Suf ↓ | AUROC ↑ | Suf ↓ | AUROC ↑ | Suf ↓ | AUROC ↑ | Suf ↓ |
| CE | 0.794 (0.000) | 0.050 (0.003) | 0.760 (0.001) | 0.093 (0.003) | **0.794** (0.000) | 0.050 (0.004) | 0.760 (0.001) | 0.093 (0.004) |
| CE + Suf-Reg | 0.792 (0.001) | 0.054 (0.001) | 0.782 (0.020) | **0.020** (0.000) | 0.781 (0.006) | 0.054 (0.002) | 0.779 (0.012) | **0.043** (0.016) |
| CE + CMI-Reg | **0.802** (0.002) | **0.043** (0.003) | **0.878** (0.006) | 0.022 (0.001) | 0.782 (0.007) | **0.025** (0.003) | **0.782** (0.038) | 0.061 (0.005) |
| Forward | **0.797** (0.001) | 0.041 (0.001) | 0.799 (0.008) | 0.080 (0.002) | **0.786** (0.006) | 0.040 (0.002) | 0.686 (0.029) | 0.098 (0.008) |
| Forward + Suf-Reg | 0.782 (0.022) | 0.041 (0.001) | 0.854 (0.009) | 0.045 (0.018) | 0.768 (0.006) | 0.048 (0.003) | 0.558 (0.266) | **0.067** (0.001) |
| Forward + CMI-Reg | 0.779 (0.007) | **0.031** (0.003) | **0.866** (0.009) | **0.040** (0.015) | 0.785 (0.011) | **0.038** (0.005) | **0.833** (0.042) | **0.053** (0.037) |
| Backward | 0.788 (0.002) | 0.048 (0.001) | 0.795 (0.009) | 0.082 (0.003) | 0.769 (0.002) | 0.045 (0.001) | 0.687 (0.028) | 0.098 (0.008) |
| Backward + Suf-Reg | **0.790** (0.002) | 0.048 (0.000) | 0.818 (0.002) | 0.073 (0.004) | 0.763 (0.001) | 0.068 (0.000) | 0.810 (0.012) | **0.013** (0.001) |
| Backward + CMI-Reg | 0.786 (0.002) | **0.043** (0.001) | **0.838** (0.005) | **0.061** (0.004) | **0.770** (0.008) | **0.039** (0.000) | **0.832** (0.012) | **0.060** (0.005) |
| DMI | **0.674** (0.122) | 0.038 (0.001) | 0.784 (0.004) | 0.098 (0.002) | 0.674 (0.149) | **0.038** (0.001) | **0.784** (0.005) | 0.098 (0.002) |
| DMI + Suf-Reg | 0.624 (0.015) | **0.037** (0.012) | **0.791** (0.000) | 0.091 (0.005) | 0.625 (0.018) | 0.038 (0.011) | 0.783 (0.000) | **0.097** (0.005) |
| DMI + CMI-Reg | 0.666 (0.164) | **0.037** (0.003) | **0.791** (0.003) | 0.095 (0.003) | 0.628 (0.017) | **0.038** (0.009) | **0.784** (0.001) | **0.097** (0.005) |
| ELR | **0.811** (0.003) | 0.045 (0.002) | 0.778 (0.000) | 0.092 (0.005) | **0.811** (0.004) | 0.045 (0.002) | 0.778 (0.000) | 0.092 (0.006) |
| ELR + Suf-Reg | 0.784 (0.001) | 0.042 (0.001) | **0.861** (0.002) | **0.033** (0.009) | 0.775 (0.002) | **0.043** (0.001) | **0.804** (0.025) | **0.014** (0.006) |
| ELR + CMI-Reg | 0.785 (0.000) | **0.036** (0.001) | 0.815 (0.010) | 0.036 (0.000) | **0.799** (0.007) | 0.039 (0.001) | 0.793 (0.015) | 0.046 (0.014) |
| GCE | 0.798 (0.001) | 0.039 (0.000) | 0.802 (0.001) | 0.087 (0.002) | **0.798** (0.002) | 0.039 (0.000) | 0.802 (0.001) | 0.087 (0.003) |
| GCE + Suf-Reg | 0.792 (0.001) | 0.052 (0.000) | 0.778 (0.054) | **0.032** (0.008) | 0.779 (0.007) | 0.052 (0.001) | 0.797 (0.045) | **0.024** (0.001) |
| GCE + CMI-Reg | **0.821** (0.001) | **0.020** (0.004) | **0.852** (0.016) | 0.053 (0.006) | 0.792 (0.005) | **0.029** (0.005) | **0.864** (0.000) | 0.043 (0.002) |
| Peer | **0.792** (0.002) | 0.043 (0.001) | 0.756 (0.009) | 0.082 (0.008) | **0.792** (0.003) | 0.043 (0.001) | 0.756 (0.011) | 0.082 (0.010) |
| Peer + Suf-Reg | 0.754 (0.045) | 0.031 (0.002) | 0.816 (0.027) | 0.028 (0.007) | 0.737 (0.017) | **0.018** (0.008) | 0.803 (0.007) | **0.066** (0.008) |
| Peer + CMI-Reg | 0.712 (0.015) | **0.019** (0.000) | **0.882** (0.005) | **0.026** (0.000) | 0.767 (0.016) | 0.036 (0.011) | **0.813** (0.005) | 0.076 (0.008) |

Table 7: Clean test-set AUROC and sufficiency gap (Suf) of our regularizers when added to off-the-shelf loss functions, for tabular datasets with synthetic **group–uniform** noise at $\eta = 0.8$ and real vs. **estimated** transition matrices. Models are selected using the *clean Brier score*. The best method within each base loss is **bolded**; the overall best non-Oracle model for each metric/dataset is additionally **underlined**.

| | Real $T_g$ | | | | Estimated $T_g$ | | | |
| | income | | adult | | income | | adult | |
| | AUROC ↑ | Suf ↓ | AUROC ↑ | Suf ↓ | AUROC ↑ | Suf ↓ | AUROC ↑ | Suf ↓ |
|---|---|---|---|---|---|---|---|---|
| CE | 0.755 (0.001) | 0.043 (0.000) | 0.476 (0.067) | 0.090 (0.011) | **0.755** (0.001) | 0.043 (0.000) | 0.476 (0.082) | 0.090 (0.014) |
| CE + Suf-Reg | 0.681 (0.021) | 0.026 (0.011) | 0.759 (0.060) | 0.051 (0.008) | 0.678 (0.020) | **0.021** (0.007) | 0.596 (0.151) | **0.086** (0.005) |
| CE + CMI-Reg | **0.767** (0.008) | **0.025** (0.008) | **0.831** (0.063) | **0.047** (0.015) | 0.669 (0.037) | 0.038 (0.027) | **0.622** (0.112) | 0.101 (0.015) |
| Forward | 0.753 (0.002) | 0.043 (0.002) | 0.473 (0.077) | 0.090 (0.012) | **0.756** (0.000) | 0.043 (0.001) | 0.469 (0.091) | 0.089 (0.016) |
| Forward + Suf-Reg | 0.710 (0.060) | **0.015** (0.001) | 0.845 (0.001) | 0.040 (0.014) | 0.639 (0.068) | 0.049 (0.009) | 0.361 (0.033) | **0.037** (0.018) |
| Forward + CMI-Reg | **0.759** (0.012) | 0.032 (0.006) | **0.889** (0.003) | **0.021** (0.005) | 0.598 (0.047) | **0.015** (0.014) | **0.500** (0.005) | 0.093 (0.004) |
| Backward | 0.755 (0.000) | 0.041 (0.002) | 0.472 (0.077) | 0.090 (0.013) | 0.684 (0.028) | 0.031 (0.002) | 0.469 (0.091) | 0.089 (0.016) |
| Backward + Suf-Reg | **0.765** (0.003) | **0.040** (0.001) | 0.680 (0.008) | **0.049** (0.002) | **0.750** (0.014) | **0.029** (0.001) | 0.407 (0.026) | **0.063** (0.003) |
| Backward + CMI-Reg | 0.756 (0.001) | **0.040** (0.001) | **0.791** (0.030) | 0.081 (0.006) | 0.653 (0.019) | 0.038 (0.002) | **0.545** (0.092) | 0.087 (0.005) |
| DMI | **0.649** (0.071) | **0.040** (0.002) | 0.411 (0.088) | 0.099 (0.008) | **0.649** (0.087) | **0.040** (0.003) | 0.411 (0.108) | 0.099 (0.010) |
| DMI + Suf-Reg | 0.585 (0.011) | 0.052 (0.003) | **0.631** (0.012) | 0.115 (0.006) | 0.589 (0.009) | 0.051 (0.008) | 0.423 (0.095) | 0.099 (0.008) |
| DMI + CMI-Reg | **0.649** (0.107) | 0.041 (0.002) | 0.411 (0.103) | **0.099** (0.010) | 0.567 (0.015) | 0.047 (0.006) | **0.443** (0.059) | **0.096** (0.006) |
| ELR | 0.755 (0.001) | 0.043 (0.001) | 0.476 (0.102) | 0.088 (0.018) | **0.755** (0.001) | **0.043** (0.001) | 0.476 (0.124) | 0.088 (0.023) |
| ELR + Suf-Reg | 0.757 (0.002) | 0.045 (0.001) | 0.658 (0.182) | **0.079** (0.025) | 0.662 (0.040) | **0.043** (0.029) | **0.597** (0.178) | **0.056** (0.031) |
| ELR + CMI-Reg | **0.759** (0.004) | **0.041** (0.001) | **0.726** (0.003) | 0.096 (0.006) | 0.671 (0.048) | 0.046 (0.028) | 0.545 (0.106) | 0.084 (0.008) |
| GCE | 0.753 (0.001) | 0.043 (0.001) | 0.472 (0.078) | 0.090 (0.013) | **0.753** (0.001) | **0.043** (0.001) | 0.472 (0.096) | 0.090 (0.016) |
| GCE + Suf-Reg | 0.756 (0.029) | 0.038 (0.006) | 0.759 (0.006) | 0.061 (0.013) | 0.640 (0.010) | 0.048 (0.006) | 0.394 (0.084) | **0.050** (0.040) |
| GCE + CMI-Reg | **0.759** (0.007) | **0.037** (0.004) | **0.892** (0.000) | **0.021** (0.002) | 0.661 (0.015) | 0.066 (0.002) | **0.569** (0.097) | 0.082 (0.017) |
| Peer | **0.755** (0.004) | 0.042 (0.002) | 0.720 (0.004) | 0.108 (0.002) | **0.755** (0.005) | 0.042 (0.003) | **0.500** (0.000) | **0.093** (0.004) |
| Peer + Suf-Reg | 0.695 (0.036) | 0.023 (0.004) | 0.771 (0.043) | **0.035** (0.010) | 0.642 (0.068) | **0.017** (0.011) | **0.500** (0.000) | **0.093** (0.004) |
| Peer + CMI-Reg | 0.699 (0.003) | **0.017** (0.001) | **0.858** (0.005) | 0.041 (0.008) | 0.640 (0.002) | 0.043 (0.005) | **0.500** (0.000) | **0.093** (0.004) |

no clearly labeled data is available, our regularizers consistently achieve lower sufficiency gaps than the base losses, and the selected models with the lowest sufficiency gaps across losses correspond our regularizers in six of eight combinations of datasets and noise types.

# E  Broader Impacts

In this work, we explored unfairness induced by label bias, and introduced regularizers so that models stay calibrated across groups with respect to the true labels. Though we evaluate on real-world data, we do not advocate for the blind deployment of ML models in any real-world safety critical setting (such as health, credit, or hiring). There are a myriad of other factors that should be taken into account before their deployment (e.g. privacy, regulation, interpretability, and socio-technical considerations). Misuse of such models could lead to real harm.

Table 8: Clean test-set AUROC and sufficiency gap (Suf) of our regularizers when added to off-the-shelf loss functions, for tabular datasets with synthetic **group–asymmetric** noise and **estimated** transition matrices. Models are selected using the *noisy Brier score*. The best method within each base loss is **bolded**; the best non-Oracle model overall for each metric/dataset is additionally **underlined**.

| | income | | adult | | lsac | | crime | |
|---|---|---|---|---|---|---|---|---|
| | AUROC ↑ | Suf ↓ | AUROC ↑ | Suf ↓ | AUROC ↑ | Suf ↓ | AUROC ↑ | Suf ↓ |
| CE | **0.682** (0.001) | **0.065** (0.002) | 0.889 (0.001) | 0.026 (0.004) | **0.561** (0.047) | 0.109 (0.006) | 0.740 (0.023) | **0.072** (0.001) |
| CE + Suf-Reg | 0.679 (0.002) | 0.076 (0.006) | 0.884 (0.003) | 0.028 (0.001) | 0.477 (0.109) | **0.092** (0.007) | 0.722 (0.042) | 0.074 (0.021) |
| CE + CMI-Reg | 0.663 (0.003) | 0.079 (0.003) | **0.892** (0.000) | **0.023** (0.002) | 0.479 (0.021) | 0.102 (0.010) | 0.731 (0.013) | 0.078 (0.010) |
| Forward | 0.633 (0.003) | 0.109 (0.004) | **0.886** (0.001) | 0.029 (0.000) | 0.446 (0.065) | 0.095 (0.001) | **0.689** (0.057) | **0.057** (0.003) |
| Forward + Suf-Reg | **0.682** (0.030) | **0.043** (0.018) | 0.884 (0.003) | 0.028 (0.001) | 0.283 (0.059) | **0.060** (0.028) | 0.596 (0.025) | 0.072 (0.009) |
| Forward + CMI-Reg | 0.609 (0.009) | 0.103 (0.007) | 0.890 (0.001) | **0.026** (0.002) | 0.554 (0.169) | 0.092 (0.019) | 0.637 (0.114) | 0.084 (0.002) |
| Backward | 0.607 (0.004) | 0.092 (0.006) | 0.886 (0.002) | 0.029 (0.001) | 0.445 (0.061) | 0.094 (0.002) | **0.564** (0.039) | **0.098** (0.013) |
| Backward + Suf-Reg | **0.612** (0.000) | **0.091** (0.003) | 0.886 (0.002) | **0.027** (0.001) | **0.506** (0.098) | **0.090** (0.001) | 0.526 (0.001) | 0.111 (0.022) |
| Backward + CMI-Reg | 0.601 (0.005) | 0.106 (0.000) | **0.887** (0.002) | 0.027 (0.000) | 0.476 (0.093) | 0.094 (0.001) | 0.563 (0.073) | 0.099 (0.009) |
| DMI | **0.590** (0.094) | 0.047 (0.013) | 0.888 (0.001) | 0.024 (0.004) | 0.477 (0.138) | 0.090 (0.001) | **0.475** (0.027) | 0.108 (0.020) |
| DMI + Suf-Reg | **0.590** (0.113) | **0.046** (0.016) | 0.856 (0.003) | 0.017 (0.002) | **0.503** (0.130) | 0.090 (0.007) | 0.459 (0.029) | 0.101 (0.037) |
| DMI + CMI-Reg | 0.587 (0.114) | **0.046** (0.017) | **0.890** (0.003) | 0.024 (0.006) | 0.480 (0.170) | **0.089** (0.001) | 0.453 (0.035) | **0.097** (0.034) |
| ELR | 0.656 (0.001) | 0.077 (0.003) | **0.887** (0.001) | 0.026 (0.001) | **0.520** (0.043) | 0.101 (0.011) | **0.656** (0.072) | **0.088** (0.002) |
| ELR + Suf-Reg | **0.677** (0.001) | **0.066** (0.002) | 0.886 (0.002) | 0.026 (0.000) | 0.349 (0.065) | **0.088** (0.011) | 0.505 (0.051) | 0.092 (0.038) |
| ELR + CMI-Reg | 0.650 (0.003) | 0.080 (0.003) | 0.883 (0.001) | **0.020** (0.002) | 0.375 (0.051) | 0.095 (0.006) | 0.650 (0.094) | 0.098 (0.007) |
| GCE | 0.675 (0.005) | **0.070** (0.000) | 0.884 (0.001) | 0.029 (0.001) | 0.491 (0.051) | 0.100 (0.001) | **0.743** (0.025) | 0.083 (0.020) |
| GCE + Suf-Reg | **0.677** (0.003) | 0.078 (0.007) | 0.884 (0.003) | 0.027 (0.001) | 0.467 (0.118) | **0.089** (0.010) | 0.719 (0.049) | **0.075** (0.027) |
| GCE + CMI-Reg | 0.663 (0.008) | 0.082 (0.007) | **0.891** (0.001) | **0.024** (0.004) | **0.623** (0.066) | 0.098 (0.000) | 0.701 (0.018) | 0.084 (0.004) |
| Peer | **0.730** (0.018) | **0.030** (0.000) | 0.888 (0.000) | 0.025 (0.001) | **0.566** (0.033) | 0.112 (0.005) | **0.696** (0.041) | **0.080** (0.001) |
| Peer + Suf-Reg | 0.644 (0.046) | 0.060 (0.037) | 0.883 (0.001) | **0.018** (0.007) | 0.280 (0.030) | **0.053** (0.007) | 0.498 (0.064) | 0.085 (0.026) |
| Peer + CMI-Reg | 0.589 (0.002) | 0.054 (0.006) | **0.892** (0.001) | 0.021 (0.001) | 0.481 (0.097) | 0.101 (0.000) | 0.663 (0.061) | 0.100 (0.011) |
| GroupPeer | – | – | **0.896** (0.002) | 0.023 (0.003) | **0.659** (0.001) | 0.094 (0.018) | – | – |
| Oracle | 0.854 (0.001) | 0.003 (0.000) | 0.919 (0.002) | 0.013 (0.001) | 0.838 (0.011) | 0.018 (0.009) | 0.825 (0.013) | 0.046 (0.010) |

Table 9: Clean test-set AUROC and sufficiency gap (Suf) of our regularizers when added to off-the-shelf loss functions, for tabular datasets with synthetic **group–uniform** noise and **estimated** transition matrices. Models are selected using the *noisy Brier score*. The best method within each base loss is **bolded**; the overall best non-Oracle model for each metric/dataset is additionally **underlined**.

| | income | | adult | | lsac | | crime | |
|---|---|---|---|---|---|---|---|---|
| | AUROC ↑ | Suf ↓ | AUROC ↑ | Suf ↓ | AUROC ↑ | Suf ↓ | AUROC ↑ | Suf ↓ |
| CE | 0.630 (0.004) | 0.053 (0.006) | **0.634** (0.019) | **0.108** (0.001) | **0.565** (0.050) | 0.111 (0.006) | 0.609 (0.015) | **0.090** (0.003) |
| CE + Suf-Reg | 0.651 (0.027) | 0.042 (0.022) | 0.573 (0.026) | 0.109 (0.029) | 0.464 (0.083) | **0.093** (0.008) | 0.647 (0.005) | 0.090 (0.020) |
| CE + CMI-Reg | **0.681** (0.003) | **0.034** (0.008) | 0.593 (0.000) | 0.112 (0.019) | 0.494 (0.036) | 0.101 (0.012) | **0.650** (0.057) | 0.092 (0.005) |
| Forward | **0.764** (0.003) | 0.047 (0.001) | **0.656** (0.009) | 0.107 (0.004) | **0.456** (0.080) | 0.095 (0.000) | 0.609 (0.014) | 0.103 (0.009) |
| Forward + Suf-Reg | 0.692 (0.005) | **0.027** (0.007) | 0.552 (0.352) | **0.013** (0.004) | 0.447 (0.131) | 0.089 (0.020) | 0.435 (0.037) | 0.102 (0.003) |
| Forward + CMI-Reg | 0.649 (0.019) | 0.034 (0.007) | 0.592 (0.010) | 0.095 (0.011) | 0.453 (0.140) | **0.087** (0.014) | **0.655** (0.068) | **0.084** (0.013) |
| Backward | 0.747 (0.003) | 0.048 (0.001) | 0.669 (0.008) | 0.106 (0.004) | 0.454 (0.074) | 0.096 (0.001) | 0.531 (0.084) | 0.094 (0.009) |
| Backward + Suf-Reg | **0.761** (0.006) | **0.047** (0.002) | **0.705** (0.014) | **0.054** (0.013) | 0.471 (0.140) | **0.085** (0.018) | 0.460 (0.003) | 0.091 (0.009) |
| Backward + CMI-Reg | 0.659 (0.008) | **0.041** (0.007) | 0.530 (0.002) | 0.096 (0.004) | **0.485** (0.100) | 0.095 (0.002) | **0.538** (0.094) | **0.089** (0.001) |
| DMI | **0.660** (0.062) | 0.034 (0.009) | **0.641** (0.065) | **0.077** (0.020) | 0.478 (0.144) | **0.088** (0.000) | 0.489 (0.043) | 0.109 (0.016) |
| DMI + Suf-Reg | 0.643 (0.090) | 0.051 (0.003) | 0.573 (0.026) | 0.078 (0.020) | 0.472 (0.171) | 0.089 (0.002) | **0.634** (0.087) | **0.081** (0.023) |
| DMI + CMI-Reg | 0.642 (0.042) | **0.025** (0.007) | 0.604 (0.020) | 0.107 (0.004) | **0.482** (0.177) | 0.088 (0.001) | 0.488 (0.054) | 0.110 (0.017) |
| ELR | 0.630 (0.001) | 0.056 (0.002) | **0.632** (0.010) | 0.112 (0.005) | **0.523** (0.042) | 0.101 (0.011) | 0.608 (0.027) | **0.088** (0.005) |
| ELR + Suf-Reg | 0.642 (0.012) | 0.043 (0.013) | 0.290 (0.026) | **0.029** (0.015) | 0.330 (0.077) | **0.083** (0.013) | 0.459 (0.049) | 0.108 (0.007) |
| ELR + CMI-Reg | **0.694** (0.001) | **0.029** (0.008) | 0.405 (0.002) | 0.066 (0.002) | 0.375 (0.052) | 0.094 (0.007) | **0.645** (0.065) | 0.096 (0.004) |
| GCE | 0.646 (0.004) | 0.052 (0.005) | **0.629** (0.004) | 0.112 (0.007) | 0.488 (0.062) | 0.097 (0.000) | 0.606 (0.016) | 0.090 (0.007) |
| GCE + Suf-Reg | **0.683** (0.046) | **0.026** (0.020) | 0.541 (0.028) | 0.124 (0.019) | 0.459 (0.086) | **0.092** (0.008) | 0.647 (0.006) | **0.086** (0.001) |
| GCE + CMI-Reg | 0.637 (0.026) | 0.043 (0.026) | 0.607 (0.000) | **0.101** (0.009) | **0.614** (0.029) | 0.096 (0.002) | **0.651** (0.068) | 0.093 (0.004) |
| Peer | **0.767** (0.000) | 0.045 (0.003) | **0.640** (0.020) | 0.107 (0.001) | **0.573** (0.040) | 0.110 (0.007) | 0.618 (0.028) | **0.085** (0.012) |
| Peer + Suf-Reg | 0.581 (0.028) | 0.042 (0.014) | 0.559 (0.282) | **0.034** (0.025) | 0.389 (0.100) | 0.095 (0.026) | 0.501 (0.074) | 0.108 (0.010) |
| Peer + CMI-Reg | 0.654 (0.007) | 0.042 (0.009) | 0.513 (0.011) | 0.092 (0.003) | 0.487 (0.097) | 0.099 (0.003) | **0.657** (0.064) | 0.096 (0.008) |
| GroupPeer | – | – | 0.287 (0.095) | 0.088 (0.003) | **0.646** (0.007) | 0.098 (0.024) | – | – |
| Oracle | 0.854 (0.001) | 0.003 (0.000) | 0.919 (0.002) | 0.013 (0.001) | 0.838 (0.011) | 0.018 (0.009) | 0.825 (0.013) | 0.046 (0.010) |

