# OpenReview forum: "On Group Sufficiency Under Label Bias"
_NeurIPS.cc/2025/Conference — NeurIPS 2025 poster_

### Official Review · Reviewer_HAo2 · 2025-06-24

**Clarity:** 2
**Significance:** 2
**Originality:** 2
**Rating:** 4
**Confidence:** 3

**Summary:**

The paper studies the problem of mitigating bias in the presence of noisy labels. It specifically considers the fairness notion of **Sufficiency**, where the true outcomes are independent of the group conditioned on the prediction. First, the authors provide a theoretical analysis demonstrating that enforcing fairness with respect to noisy sensitive attributes necessarily results in a non-negative group sufficiency gap with respect to the true labels. They then propose a regularization based on mutual information that can upper bound the sufficiency gap, assuming access to noisy labels and a transition probability matrix $P(y_{\text{noise}} \mid y, \text{g})$. The paper also presents an extensive experimental study on multiple datasets with both synthetic and real noisy data to evaluate their proposed regulation method.

**Questions:**

**Q1.** My first question concerns the entropy term in Equation (4). Specifically, the entropy $H(Y \mid F(X))$ is not conditioned on the sensitive attribute $G$, so the use of  $T_{g_i}$ is a bit confusing to me. This also appears in the appendix (lines 1049–1055), particularly in line 1052, where the definition of mutual information appears to differ from the one presented in line 202 of the main text. Could you clarify this discrepancy?

The rest of my questions concern the experiments.
**Q2.** Regarding the "Noise Types," I would like to ask why the authors add noise only to one specific sensitive group. Performing additional experiments with the same level of noise applied to different groups seems straightforward and would provide a more comprehensive evaluation.

**Q3.** Regarding the additional experiments, as mentioned before, further evaluation—especially studying the effect of different noise levels in the dataset—would be beneficial. Combining this with experiments that use estimated transition probabilities would provide a more thorough evaluation.

**Ethical Concerns:**

["NO or VERY MINOR ethics concerns only"]

**Final Justification:**

The paper is strong overall, with its main weakness lying in the presentation of the core concepts and the experimental evaluation.

While the authors clearly explain their motivation behind the proposed formulation, I believe the paper could benefit from a clearer presentation of the backward correction approach and the role of the regularization.

Regarding the experiments, I would have expected a more thorough evaluation in the main part of the paper, including:
1. Results under different noise levels (also small noise levels),
2. A comparison between the estimated and true transition matrices,
along with a more conclusive analysis of the impact of both regularization methods.

**Limitations:**

yes

**Quality:**

2

**Strengths And Weaknesses:**

**Strengths**
1. The paper is well written, easy to follow, and presents a comprehensive literature review.
2. The concept of noisy labels is particularly interesting and effectively captures many practical challenges.
3. The authors contribute by focusing on the notion of *sufficiency*, which has received less attention compared to other fairness notions in the literature.
4. They provide a theoretical analysis that highlights the importance of their proposed regularization approach.
5. Extensive experiments are conducted to evaluate the performance of the method.

**Weaknesses**
1. **Dependence on transition probabilities**: The method requires knowledge of the transition probabilities. While the authors acknowledge this limitation, and I understand that estimating such quantities may require additional information, there are existing methods that attempt to relax this assumption (e.g., reference [48]). The claim in line 126 that the method requires only noisy labels seems overstated, as the transition probabilities are also needed.

2.  **Entropy term and conditioning**: My second concern is about the correctness of the proposed regularization method. I found it somewhat difficult to follow the derivation of the entropy term in Equation (4), particularly the use of \( T_{g_i} \) as the entropy is not conditioning to the sensitive group. See also question Q1.

3. **Evaluation**: Although the paper includes many experiments, additional evaluation—especially studying the effect of different noise levels in the dataset—would be beneficial.

---

> ### Author Rebuttal · Authors · 2025-07-31
>
> Thank you for the insightful review and constructive feedback!
>
> > W1: Dependence on transition probabilities. The claim in line 126 that the method requires only noisy labels seems overstated, as the transition probabilities are also needed.
>
>
> We note that we do experiment with using Confident Learning [1] to estimate the transition matrix only from noisy data, though this generally results in worse performance than using the true transition matrix. We emphasize that as better transition matrix estimates are developed, our method will improve in this setting as well. We will clarify the claim in L126 to reflect this.
>
>
>
> > W2: Entropy term and conditioning.
>
> > Q1: My first question concerns the entropy term in Equation (4). Specifically, the entropy H(Y | F(X))  is not conditioned on the sensitive attribute G, so the use of T_{g_i} is a bit confusing to me. In line 1052, where the definition of mutual information appears to differ from the one presented in line 202 of the main text. Could you clarify this discrepancy?
>
>
> In our class-attribute-conditioned noise model (Assumption 3.1), the backward correction always involves $T_g$ because noise depends on $(Y, G)$, which leads to $\\mathbb{E}_{\\tilde Y \\mid Y = y,\\, G = g} [(T_g^{-1})\_{\\tilde{Y},\\cdot}] = e\_y^{\\top}$. Using the marginal matrix $T$ would result in a biased estimator, as it does not correspond to the actual probability of corruption for that sample.
>
> After this backwards correction, $−\\log \\pi(Y \\mid f(X)) $ is independent of $G$. Thus, the two definitions on L1052 and L202 are equivalent. We will modify L1050 to the following to make it more explicit:
>
> $\\mathbb{E}\_{\\tilde Y \\mid G, Y=y}[(-T_G^{-1}\\log\pi(\\cdot\\mid s))\_{\\tilde Y}] = -\\log\\pi( y\\mid s)$.
>
>
>
>
> > Q2: Regarding the "Noise Types," I would like to ask why the authors add noise only to one specific sensitive group. Performing additional experiments with the same level of noise applied to different groups seems straightforward and would provide a more comprehensive evaluation.
>
> We add noise only to one specific group as it is the worst-case scenario. Because the entire disparity is concentrated in one group, the resulting sufficiency gap is maximal for a fixed average noise rate. Further, if both groups were noisy, as one group's noise matrix approaches the other, training a naive calibrated classifier with respect to noisy labels is sufficient to remove the sufficiency gap as shown in Corollary 4.1.1, and thus the problem becomes trivial.
>
> We appreciate the suggestion to add noise _solely_ to the _other_ protected group. In our current experiments, the noisy group was selected largely for historical reasons, and we did not experiment with noising the other group due to computational considerations. We expect our method to perform similarly with noise in the other group, and will verify this in a future revision.
>
>
>
>
>
>
>
> [1] Confident Learning: Estimating Uncertainty in Dataset Labels. JAIR.

---

> > ### Author Response · Authors · 2025-07-31
> >
> > > W3: Evaluation. Although the paper includes many experiments, additional evaluation—especially studying the effect of different noise levels in the dataset—would be beneficial.
> >
> > > Q3. Regarding the additional experiments, as mentioned before, further evaluation—especially studying the effect of different noise levels in the dataset—would be beneficial. Combining this with experiments that use estimated transition probabilities would provide a more thorough evaluation.
> >
> >
> > Thank you for the suggestion! Due to limited computational resources, we were able to run our method for one additional noise level ($\\eta = 0.6$, compared to $\\eta = 0.75$ in the paper), for two datasets and for both real and estimated transition matrices. We find that CMI-REG gives a lower sufficiency gap for every single loss type for both datasets. Here, using an estimated transition matrix does not seem to degrade performance much, and we empirically find that the estimated transition matrix is closer to the true one than when $\\eta = 0.75$. We will finish this experiment for the remaining datasets and add it to the appendices in the revision.
> >
> >
> >
> > |                    |               | Real $T_g$            |               |                   |               | Estimated $T_g$       |               |                   |
> > |--------------------|---------------|-------------------|---------------|-------------------|---------------|-------------------|---------------|-------------------|
> > |                    |        income |                   |         adult |                   |        income |                   |         adult |                   |
> > |                    |         AUROC |               Suf |         AUROC |               Suf |         AUROC |               Suf |         AUROC |               Suf |
> > |                 CE | 0.794 (0.000) |     0.050 (0.003) | 0.760 (0.001) |     0.093 (0.003) | 0.794 (0.000) |     0.050 (0.004) | 0.760 (0.001) |     0.093 (0.004) |
> > |       CE + CMI-Reg | 0.802 (0.002) | **0.043 (0.003)** | 0.878 (0.006) | **0.022 (0.001)** | 0.782 (0.007) | **0.025 (0.003)** | 0.782 (0.038) | **0.061 (0.005)** |
> > |            Forward | 0.797 (0.001) |     0.041 (0.001) | 0.799 (0.008) |     0.080 (0.002) | 0.786 (0.006) |     0.040 (0.002) | 0.686 (0.029) |     0.098 (0.008) |
> > | Forward + CMI-Reg  | 0.779 (0.007) | **0.031 (0.003)** | 0.866 (0.009) | **0.040 (0.015)** | 0.785 (0.011) | **0.038 (0.005)** | 0.833 (0.042) | **0.053 (0.037)** |
> > |           Backward | 0.788 (0.002) |     0.048 (0.001) | 0.795 (0.009) |     0.082 (0.003) | 0.769 (0.002) |     0.045 (0.001) | 0.687 (0.028) |     0.098 (0.008) |
> > | Backward + CMI-Reg | 0.786 (0.002) | **0.043 (0.001)** | 0.838 (0.005) | **0.061 (0.004)** | 0.770 (0.008) | **0.039 (0.000)** | 0.832 (0.012) | **0.060 (0.005)** |
> > |                DMI | 0.674 (0.122) |     0.038 (0.001) | 0.784 (0.004) |     0.098 (0.002) | 0.674 (0.149) | **0.038 (0.001)** | 0.784 (0.005) |     0.098 (0.002) |
> > |      DMI + CMI-Reg | 0.666 (0.164) | **0.037 (0.003)** | 0.791 (0.003) | **0.095 (0.003)** | 0.628 (0.017) | **0.038 (0.009)** | 0.784 (0.001) | **0.097 (0.005)** |
> > |                ELR | 0.811 (0.003) |     0.045 (0.002) | 0.778 (0.000) |     0.092 (0.005) | 0.811 (0.004) |     0.045 (0.002) | 0.778 (0.000) |     0.092 (0.006) |
> > |      ELR + CMI-Reg | 0.785 (0.000) | **0.036 (0.001)** | 0.815 (0.010) | **0.036 (0.000)** | 0.799 (0.007) | **0.039 (0.001)** | 0.793 (0.015) | **0.046 (0.014)** |
> > |                GCE | 0.798 (0.001) |     0.039 (0.000) | 0.802 (0.001) |     0.087 (0.002) | 0.798 (0.002) |     0.039 (0.000) | 0.802 (0.001) |     0.087 (0.003) |
> > |      GCE + CMI-Reg | 0.821 (0.001) | **0.020 (0.004)** | 0.852 (0.016) | **0.053 (0.006)** | 0.798 (0.007) | **0.031 (0.002)** | 0.864 (0.000) | **0.043 (0.002)** |
> > |               Peer | 0.792 (0.002) |     0.043 (0.001) | 0.756 (0.009) |     0.082 (0.008) | 0.792 (0.003) |     0.043 (0.001) | 0.756 (0.011) |     0.082 (0.010) |
> > |     Peer + CMI-Reg | 0.712 (0.015) | **0.019 (0.000)** | 0.882 (0.005) | **0.026 (0.000)** | 0.767 (0.016) | **0.036 (0.011)** | 0.813 (0.005) | **0.076 (0.008)** |

---

> > ### Comment · Reviewer_HAo2 · 2025-08-01
> > **Additional clarification of Q1.**
> >
> > Dear Authors, thank you for your response.
> >
> > I appreciate the effort you put into addressing the questions.
> >
> > I am still a bit confused by your answer to question **Q1**.
> > Could you please elaborate further on how exactly Eq (4) is derived?

---

> > > ### Author Response · Authors · 2025-08-02
> > > **Clarification of Equation (4)**
> > >
> > > Thank you for the quick response! Here is a re-derivation and proof showing the unbiasedness of Equation (4) which mostly follows the proof of the backwards correction from [1] for our specific setup, but provides more exposition.
> > >
> > > Let $\\mathbf{h}\_Y(\\cdot \\mid s\_i) := [-\\log p\_Y(1 \\mid s\_i), ..., -\\log p\_Y(K \\mid s\_i)] $, and $\\mathbf{h}\_{\\tilde{Y}}(\\cdot \\mid s\_i)$ similarly. Let $[\\cdot]\_{\\tilde{y}\_i}$ denote taking the $\\tilde{y}\_i$-th element of this $K$-dimensional vector.
> > >
> > > First, we start with the definition of entropy:
> > >
> > > $H(Y \\mid f(X)) = \\mathbb{E}\_{X, Y}[-\\log p_Y(Y \\mid f(X))]$,
> > >
> > > which we can estimate given a minibatch $\\{(x\_i, g\_i, \\tilde{y}\_i)\_{i=1}^B\\}$ by applying the (unbiased) Monte Carlo estimator, defining $s\_i = f(x\_i)$ for convenience:
> > >
> > > $\\hat{H}(Y \\mid f(X)) = \\frac{1}{B} \\sum\_{i=1}^B -\\log p\_Y(y\_i \\mid s\_i) = \\frac{1}{B} \\sum\_{i=1}^B [\\mathbf{h}\_Y(\\cdot \\mid s\_i)]\_{y\_i}$.
> > >
> > >
> > > Since we do not observe $y\_i$, we propose instead to use the backward-correction estimator shown in Equation (4):
> > >
> > > $\\hat{H}'(Y \\mid f(X)) = \\frac{1}{B} \\sum\_{i=1}^B [T\_{g\_i}^{-1}\\,\\mathbf{h}\_Y(\\cdot \\mid s\_i)]\_{\\tilde{y}\_i}$.
> > >
> > > We want to show that this estimator is unbiased, i.e. $\\mathbb{E}\_{X,G,Y,\\tilde{Y}}[\\hat{H}'(Y \\mid f(X))] = H(Y \\mid f(X))$.
> > >
> > > **Proof.**
> > >
> > > First, observe that
> > >
> > > \\[
> > > \\begin{aligned}
> > > \\mathbb{E}\_{\\tilde{Y}\\mid Y=y\_i,\\,G=g\_i}
> > >    \\bigl[\\bigl(T\_{g\_i}^{-1}\\mathbf{h}\_Y(\\cdot\\mid s\_i)\\bigr)\_{\\tilde Y}\\bigr]
> > >   &= \\sum\_{c=1}^K \\mathbb{P}(\\tilde Y=c\\mid Y=y\_i,G=g\_i)\\;
> > >      \\bigl[T\_{g\_i}^{-1}\\mathbf{h}\_Y(\\cdot\\mid s\_i)\\bigr]\_c \\\\[4pt]
> > >   &= \\sum\_{c=1}^K T^{g\_i}\_{y\_i,c}\\;
> > >      \\bigl[T\_{g\_i}^{-1}\\mathbf{h}\_Y(\\cdot\\mid s\_i)\\bigr]\_c \\\\[4pt]
> > >   &= e\_{y\_i}^{\\top}\\,T\_{g\_i}\\,T\_{g\_i}^{-1}\\,\\mathbf{h}\_Y(\\cdot\\mid s\_i) \\\\[4pt]
> > >   &= \\bigl[\\mathbf{h}\_Y(\\cdot\\mid s\_i)\\bigr]\_{y\_i} \\\\[4pt]
> > >   &= -\\log p\_Y(y\_i\\mid s\_i). & (1)
> > > \\end{aligned}
> > > \\]
> > >
> > > which is no longer a function of $g\_i$. Similarly,
> > > \\[
> > > \\mathbb{E}\_{\\tilde{Y}\\mid Y, G}\\bigl[\\bigl(T\_{G}^{-1}\\mathbf{h}\_Y(\\cdot\\mid s\_i)\\bigr)\_{\\tilde Y}\\bigr]
> > >   = -\\log p\_Y(Y \\mid s\_i).
> > > \\]
> > >
> > > Then,
> > > \\[
> > > \\begin{aligned}
> > > \\mathbb{E}\_{X,G,Y,\\tilde{Y}}[\\hat{H}'(Y \\mid f(X))]
> > >   &= \\mathbb{E}\_{X,G,Y}\\bigl[\\mathbb{E}\_{\\tilde{Y}\\mid X,G,Y}[\\hat{H}'(Y \\mid f(X))]\\bigr] \\\\
> > >   &= \\mathbb{E}\_{X,G,Y}\\bigl[\\mathbb{E}\_{\\tilde{Y}\\mid G,Y}[\\hat{H}'(Y \\mid f(X))]\\bigr] & \\text{(Assumption 3.1)} \\\\
> > >   &= \\mathbb{E}\_{X,G,Y}[-\\log p\_Y(Y \\mid f(X))] & \\text{(Substituting (1))} \\\\
> > >   &= \\mathbb{E}\_{X,Y}[-\\log p\_Y(Y \\mid f(X))] \\\\
> > >   &= H(Y \\mid f(X)).
> > > \\end{aligned}
> > > \\]
> > >
> > > Thus, $\\hat{H}'$ is an unbiased estimator of $H(Y \\mid f(X))$.
> > >
> > >
> > > Please let us know if this addresses your concerns! We will update the proof of Lemma A.5 with this clearer version.
> > >
> > >
> > > [1] Making Deep Neural Networks Robust to Label Noise: a Loss Correction Approach. CVPR 2017.

---

> ### Comment · Reviewer_HAo2 · 2025-08-04
>
> Dear Authors, thank you for your detailed response.
>
> Your response seems satisfactory; I still do not understand how, in the backward correction, you are allowed to use $T_g$ instead of $T$, as the definition of entropy does not depend on $g_i$.

---

> ### Author Response · Authors · 2025-08-04
>
> While it's true that the true entropy
>
> $H(Y \\mid f(X)) = \\mathbb{E}\_{X, Y}[-\\log p_Y(Y \\mid f(X))]$
>
> does not explicitly depend on $G$, the _estimator we construct for this quantity_ does. This is because we only observe $\tilde{Y}$  and not $Y$, and given Assumption 3.1, we need $T_g$ to invert the noise generating process to "recover" $Y$. By using $T_g$ in the estimator, we achieve unbiasedness as is shown above. One could instead use $T$ in the estimator, but it would be biased. We will make this distinction clearer in the revision. Please let us know if this addresses your concerns!

---

> ### Comment · Reviewer_HAo2 · 2025-08-06
>
> Dear Authors, thank you for your detailed reply.
>
> I understand your point and will increase my score. I believe the paper would benefit from a more detailed explanation, as the result currently appears counterintuitive.
>
> Additionally, regarding the experimental section, I would suggest presenting results with different noise levels (beyond 0.6 and 0.7) and comparing the performance using the true versus the estimated transition matrix for both CMI-REG and SUF-REG in the main part of the paper. This would offer a clearer picture of the method’s effectiveness.

---

> > ### Author Response · Authors · 2025-08-06
> >
> > We are glad to hear that your concerns have been addressed, and we sincerely appreciate your updated evaluation of the paper!
> >
> > > I understand your point and will increase my score. I believe the paper would benefit from a more detailed explanation, as the result currently appears counterintuitive.
> >
> > We will add the above explanation to Section 5 of the main text, and to the proof of Lemma A.5.
> >
> > > Additionally, regarding the experimental section, I would suggest presenting results with different noise levels (beyond 0.6 and 0.7) and comparing the performance using the true versus the estimated transition matrix for both CMI-REG and SUF-REG in the main part of the paper. This would offer a clearer picture of the method’s effectiveness.
> >
> > Given the additional page we will have for the camera ready if the paper is accepted, we will add the experimental results from our response to Reviewer HAo2 (different noise level with both real and estimated $T_g$), in addition to results for one additional noise level, to the main text.
> >
> > Thank you again for the constructive feedback!

---

### Official Review · Reviewer_NBv5 · 2025-06-24

**Clarity:** 3
**Significance:** 2
**Originality:** 3
**Rating:** 5
**Confidence:** 3

**Summary:**

The paper investigates the challenge of sufficiency in datasets that contain label bias. Similar to previous work that focused on demographic parity and equalized odds the paper tackles the problem first from a theoretical point of view. The authors introduce the sufficiency gap as way of measuring the discrepancy between the sufficiency measure on the biased and unbiased labels. From this they introduce two methodologies to calculate the sufficiency gap without having access to the unbiased labels, which can be used to introduce an additional regularization term to the training loss in order to reduce this sufficiency gap.

**Questions:**

If the camera-ready allows for an additional page then I would ask the authors to increase the size of the plots in Figure 2. Pareto-fronts are still the standard and easy to interpret model capabilities. For the labels I would suggest to replace l_pred with naïve model as I assume this is the model with no bias mitigation method effecting it. It is also quite confusing how this model can exhibit a curve. It is better to plot points that have the same hyperparameters but run for several seeds be 1 point that has a confidence ellipse around it. This shows volatility of a method better. Also if the plots are larger I would suggest having the x-axis to be the same for all plots on the same dataset as this is better visualization for the reader.

The paper would also improve if the authors could provide results comparing their method with other methods, however I can see how this would fall out of the scope of the rebuttal. While I appreciate it if the authors would do this, I do not expect it.

**Ethical Concerns:**

["NO or VERY MINOR ethics concerns only"]

**Final Justification:**

My opinion of the paper remained unchanged based on the rebuttal provided by the authors. The paper works on a rather niche but interesting topic and the problem is tackled from a good theoretical point of view and validated by experiments.

**Limitations:**

Yes

**Paper Formatting Concerns:**

No concerns

**Quality:**

3

**Strengths And Weaknesses:**

The paper starts out from a strong theoretical basis. In order to come to these theorems, certain assumptions are needed. These assumptions are logically sound and do not effect the usability of the method and are in line with the promises that the paper made in the beginning with relation to the capability of the method.

The methods are tested on a wide set of datasets with different data types such as tabular, image, and text. All information is provided in the appendix of the paper in order to reproduce the experiments conducted in the paper.

The value of the novel methods proposed in this paper would have been greatly enhanced if the authors would have used a different method to compare their method against. Fairret is an example of a method that can function for predictive parity (the simplification of sufficiency when dealing with binary classification) and also works as an added loss term. These types of method would allow the authors to show that their specific focus on label bias is warranted as this cannot by done by traditional methods not assuming this level of label bias.

The paper uses the adult dataset, while still popular in the research area of fairness in AI, it has been shown that this dataset should not be used anymore for showing the capabilities of methods (Fabris et al., Algorithmic Fairness Datasets: the Story so Far). The paper contains datasets with both simulated and real noise with regards to the labels. This is already commendable but the authors could have also used the dataset of Lenders and Calders (Real-life Performance of Fairness Interventions-Introducing A New Benchmarking Dataset for Fair ML) which provides a dataset with these biased and unbiased labels on a dataset where biases are more inherently present. The dataset however is very small.

The interpretability of figure 2 could be improved; the plots are very small and the legend is not very easy to understand without reading the entire paper.

---

> ### Author Rebuttal · Authors · 2025-07-31
>
> Thank you for the insightful review and constructive feedback!
>
> > The value of the novel methods proposed in this paper would have been greatly enhanced if the authors would have used a different method to compare their method against. Fairret is an example of a method that can function for predictive parity (the simplification of sufficiency when dealing with binary classification) and also works as an added loss term. These types of method would allow the authors to show that their specific focus on label bias is warranted as this cannot by done by traditional methods not assuming this level of label bias.
>
> > The paper would also improve if the authors could provide results comparing their method with other methods, however I can see how this would fall out of the scope of the rebuttal. While I appreciate it if the authors would do this, I do not expect it.
>
> We have added a multicalibration baseline from [1] following a suggestion from Reviewer xUeM. Specifically, we pre-process the data using the $T_g^{-1}$ matrix, learn an ERM model, then apply Algorithm 1 from [1] for multicalibration, using the single attribute to define $\mathcal{C}$.
>
> We present results for AUROC and Sufficiency Gap below, using the same setting as Table 2 of the main paper. We include only the Forward loss for our methods for brevity. We find that sufficiency gaps for this baseline remain substantially higher and AUROC is also significantly worse.
>
>
>
> |                   |         |                 income  |          |                 adult  |           |              lsac     |         |                crime    |
> |-------------------|--------------:|------------------:|--------------:|------------------:|--------------:|------------------:|--------------:|------------------:|
> |                   |         AUROC |               Suf |         AUROC |               Suf |         AUROC |               Suf |         AUROC |               Suf |
> |           Forward | 0.801 (0.002) | **0.038 (0.002)** | 0.887 (0.001) | **0.024 (0.004)** | 0.436 (0.006) |     0.097 (0.001) | 0.788 (0.011) |     0.056 (0.010) |
> | Forward + Suf-Reg | 0.812 (0.002) |     0.059 (0.003) | 0.898 (0.000) |     0.032 (0.001) | 0.690 (0.011) |     0.091 (0.016) | 0.789 (0.006) |     0.053 (0.006) |
> | Forward + CMI-Reg | 0.801 (0.002) | **0.038 (0.001)** | 0.888 (0.002) | **0.024 (0.004)** | 0.704 (0.172) | **0.065 (0.070)** | 0.802 (0.015) | **0.046 (0.011)** |
> |         GroupPeer |             - |                 - | 0.898 (0.001) |     0.035 (0.004) | 0.664 (0.010) |     0.095 (0.023) |             - |                 - |
> |                Multicalibration | 0.561 (0.019) |     0.078 (0.010) | 0.787 (0.004) |     0.017 (0.001) | 0.500 (0.000) |     0.107 (0.007) | 	0.508 (0.013) |     0.104 (0.012) |
> |            Oracle | 0.854 (0.001) |     0.003 (0.000) | 0.919 (0.002) |     0.013 (0.001) | 0.838 (0.011) |     0.018 (0.009) | 0.825 (0.013) |     0.046 (0.010) |
>
>
>
>
>
>
> > The paper uses the adult dataset, while still popular in the research area of fairness in AI, it has been shown that this dataset should not be used anymore for showing the capabilities of methods (Fabris et al., Algorithmic Fairness Datasets: the Story so Far).
>
>
> We agree with the issues with the adult dataset, and have included it here solely due to its popularity in other fairness literature. However, we have actually also conducted experiments on the ACS Income dataset from [2], which has been proposed as a drop-in replacement for adult which addresses some of its issues (age, arbitrary threshold, outdated feature encodings).
>
>
> > The paper contains datasets with both simulated and real noise with regards to the labels. This is already commendable but the authors could have also used the dataset of Lenders and Calders (Real-life Performance of Fairness Interventions-Introducing A New Benchmarking Dataset for Fair ML) which provides a dataset with these biased and unbiased labels on a dataset where biases are more inherently present. The dataset however is very small.
>
> Thank you for pointing out this dataset! We have conducted an experiment on it, using the biased human labels from the _ranking_ strategy as observed labels, and the eight categorical features from the Kaggle version of the dataset.
>
> Due to small sample size (n=856), and the fact that even the oracle (training on clean labels) has quite high Sufficiency Gap, we expect the gain of our method in this setting to be limited. However, we do find that (1) in all cases, applying CMI-Reg reduces the sufficiency gap relative to the base loss, and (2) applying CMI-Reg does not significantly change overall AUROC when the confidence intervals are taken into account.
>
>
> |             Method |         AUROC |               Suf |
> |-------------------:|--------------:|------------------:|
> |                 CE | 0.681 (0.043) |     0.050 (0.028) |
> |       CE + Suf-Reg | 0.682 (0.055) | **0.047 (0.008)** |
> |       CE + CMI-Reg | 0.683 (0.049) | **0.047 (0.030)** |
> |            Forward | 0.689 (0.048) |     0.055 (0.021) |
> |  Forward + Suf-Reg | 0.688 (0.060) | **0.052 (0.025)** |
> |  Forward + CMI-Reg | 0.674 (0.044) |     0.054 (0.014) |
> |           Backward | 0.692 (0.032) |     0.078 (0.002) |
> | Backward + Suf-Reg | 0.691 (0.053) |     0.063 (0.016) |
> | Backward + CMI-Reg | 0.688 (0.058) | **0.051 (0.007)** |
> |                DMI | 0.688 (0.044) |     0.061 (0.005) |
> |      DMI + Suf-Reg | 0.688 (0.056) |     0.051 (0.008) |
> |      DMI + CMI-Reg | 0.708 (0.036) | **0.048 (0.003)** |
> |                ELR | 0.667 (0.035) |     0.060 (0.012) |
> |      ELR + Suf-Reg | 0.669 (0.043) |     0.067 (0.022) |
> |      ELR + CMI-Reg | 0.666 (0.040) | **0.057 (0.009)** |
> |                GCE | 0.683 (0.026) |     0.073 (0.017) |
> |      GCE + Suf-Reg | 0.688 (0.056) |     0.054 (0.018) |
> |      GCE + CMI-Reg | 0.681 (0.053) | **0.048 (0.009)** |
> |               Peer | 0.681 (0.035) |     0.067 (0.020) |
> |     Peer + Suf-Reg | 0.660 (0.006) |     0.062 (0.011) |
> |     Peer + CMI-Reg | 0.692 (0.043) | **0.053 (0.018)** |
> |          GroupPeer | 0.687 (0.046) |     0.069 (0.012) |
> |             Oracle | 0.713 (0.075) |     0.050 (0.009) |
>
>
>
> > The interpretability of figure 2 could be improved; the plots are very small and the legend is not very easy to understand without reading the entire paper.
>
> > If the camera-ready allows for an additional page then I would ask the authors to increase the size of the plots in Figure 2.
>
> Thank you for these suggestions! We will make these changes for the next revision if additional space is available.
>
>
> [1] Multicalibration: Calibration for the (Computationally-Identifiable) Masses. ICML 2018.
>
> [2] Retiring Adult: New Datasets for Fair Machine Learning. NeurIPS 2021.

---

> > ### Comment · Reviewer_NBv5 · 2025-08-04
> >
> > I would like to thank the authors for their efforts in conducting novel experiments based on the comments they received in both mine and the other reviews. As my score was already positive, I do not see to change my score based on the rebuttal.

---

### Official Review · Reviewer_fbDW · 2025-07-02

**Clarity:** 2
**Significance:** 3
**Originality:** 3
**Rating:** 4
**Confidence:** 4

**Summary:**

This paper studies the group sufficiency fairness metric in the presence of group-dependent label noise. The authors propose a regularization-based fix for such label noise and perform well-designed experiments to demonstrate the effectiveness of their approach.

**Questions:**

I like the problem, but the two severe limitations overpower the theoretical analysis performed by the authors. I would want to ask the authors the following to reconsider the rating on this paper:
1. In the noisiest of settings where you can only estimate Brier scores and transition matrices with noisy data (perhaps with the help of Clusterability or some anchor points), will the proposed CMI-REG show a lot of significant improvement?
2. Can the authors comment on the convergence of the proposed mutual information estimators with finite samples? Off the top of the head, they can either argue that it minimizes a relaxed version of the sufficiency gap or assume a parametric distribution on data (Gaussian) to show that it at least works as intended on those kinds of distributions.

**Ethical Concerns:**

["NO or VERY MINOR ethics concerns only"]

**Final Justification:**

The authors have provided results for the setting provided and even managed to supply some analysis for their estimator.

**Limitations:**

yes

**Paper Formatting Concerns:**

No formatting concerns.

**Quality:**

3

**Strengths And Weaknesses:**

Strengths:

1. The problem of studying group sufficiency with label noise surprisingly flew under the radar. Many fairness metrics have been studied with label bias except this one, and thus, this work identifies an important gap.
2. Theorem 4.1 is interesting as it precisely gives how worse can sufficiency gap can get with label noise.
3. The experiments are comprehensive (however, there are some doubts, which we will discuss next).

Limitations:
1. The proposed method has some dominance over others (like vanilla noise correction, since unconstrained clean ERM is sufficient to ensure group sufficiency according to Liu et al. 2019). However, a case where the method can shine out and establish clear dominance is when the transition matrix is estimated and the Brier score itself is noisy. I couldn't find that set of results anywhere (please do point me to them if I have missed those). The supplementary does show real-world label noise results (Table 5), and the CMI-REG doesn't seem to beat vanilla CE, both in Suf-gap and Acc. Because of this, I am skeptical of the empirical advantages
2. I am also skeptical of the finite sample estimator of the CMI regularizer. Since the regularizer depends on mutual information and conditional entropy, and estimators for both of them have been known to be very tricky and have convergence issues with finite samples,  I am not sure whether the loss function would do its intended job. Had the experimental results outperformed everything else, I could have believed that the estimators are doing their job, but due to the last point, I am not sure.

---

> ### Author Rebuttal · Authors · 2025-07-31
>
> Thank you for the insightful review and constructive feedback!
>
> > The proposed method has some dominance over others. However, a case where the method can shine out and establish clear dominance is when the transition matrix is estimated and the Brier score itself is noisy.
>
> > In the noisiest of settings where you can only estimate Brier scores and transition matrices with noisy data (perhaps with the help of Clusterability or some anchor points), will the proposed CMI-REG show a lot of significant improvement?
>
> Thank you for the suggestion! We note that the setting where we only have estimated transition matrices, as well as no clean examples for model selection, is the most difficult setting for our problem. We have provided results for this setting below. We observe that across the 24 combinations of losses and datasets, CMI-REG achieves lower or equal sufficiency gap than the base loss in 20 such combinations.
>
> Further, expecting our regularizer to improve every single base loss is a lofty goal, as some losses may have optimization landscapes that naturally work better with our regularizer. Instead, looking at the best achievable sufficiency gap for each dataset, we find that CMI-REG achieves the lowest sufficiency gap across all four datasets.
>
>
>
>
>
>
> |                    |               |            income |               |             adult |               |              lsac |               |             crime |
> |--------------------|---------------|------------------:|---------------|------------------:|---------------|------------------:|---------------|------------------:|
> |                    |         AUROC |               Suf |         AUROC |               Suf |         AUROC |               Suf |         AUROC |               Suf |
> |                 CE | 0.631 (0.005) |     0.052 (0.009) | 0.634 (0.023) | **0.108 (0.001)** | 0.565 (0.061) |     0.111 (0.008) | 0.609 (0.018) |     0.090 (0.003) |
> |       CE + CMI-Reg | 0.681 (0.003) | **0.034 (0.008)** | 0.593 (0.000) |     0.112 (0.019) | 0.494 (0.036) | **0.101 (0.012)** | 0.650 (0.057) | **0.092 (0.005)** |
> |            Forward | 0.764 (0.003) |     0.047 (0.001) | 0.656 (0.012) |     0.107 (0.005) | 0.456 (0.097) |     0.095 (0.000) | 0.609 (0.018) |     0.103 (0.011) |
> | Forward + CMI-Reg  | 0.649 (0.019) | **0.034 (0.007)** | 0.592 (0.010) | **0.095 (0.011)** | 0.453 (0.140) | **0.087 (0.014)** | 0.655 (0.068) | **0.084 (0.013)** |
> |           Backward | 0.747 (0.004) |     0.048 (0.001) | 0.669 (0.010) |     0.106 (0.005) | 0.454 (0.091) |     0.096 (0.001) | 0.531 (0.103) |     0.094 (0.011) |
> | Backward + CMI-Reg | 0.659 (0.008) | **0.041 (0.007)** | 0.530 (0.002) | **0.096 (0.004)** | 0.485 (0.100) | **0.095 (0.002)** | 0.538 (0.094) | **0.089 (0.001)** |
> |                DMI | 0.660 (0.076) |     0.034 (0.011) | 0.641 (0.079) | **0.077 (0.025)** | 0.478 (0.177) | **0.088 (0.000)** | 0.437 (0.172) |     0.095 (0.014) |
> |      DMI + CMI-Reg | 0.642 (0.042) | **0.025 (0.007)** | 0.604 (0.020) |     0.107 (0.004) | 0.482 (0.177) | **0.088 (0.001)** | 0.435 (0.173) | **0.084 (0.008)** |
> |                ELR | 0.630 (0.001) |     0.056 (0.002) | 0.632 (0.012) |     0.112 (0.006) | 0.523 (0.051) |     0.101 (0.014) | 0.608 (0.033) | **0.088 (0.006)** |
> |      ELR + CMI-Reg | 0.694 (0.001) | **0.029 (0.008)** | 0.405 (0.002) | **0.066 (0.002)** | 0.375 (0.052) | **0.094 (0.007)** | 0.645 (0.065) |     0.096 (0.004) |
> |                GCE | 0.646 (0.005) |     0.052 (0.005) | 0.629 (0.004) |     0.112 (0.008) | 0.488 (0.076) |     0.097 (0.000) | 0.606 (0.020) | **0.090 (0.008)** |
> |      GCE + CMI-Reg | 0.637 (0.026) | **0.043 (0.026)** | 0.607 (0.000) | **0.101 (0.009)** | 0.614 (0.029) | **0.096 (0.002)** | 0.651 (0.068) |     0.093 (0.004) |
> |               Peer | 0.767 (0.000) |     0.045 (0.004) | 0.640 (0.024) |     0.107 (0.001) | 0.573 (0.049) |     0.110 (0.009) | 0.618 (0.034) | **0.085 (0.015)** |
> |     Peer + CMI-Reg | 0.654 (0.007) | **0.042 (0.009)** | 0.513 (0.011) | **0.092 (0.003)** | 0.487 (0.097) | **0.099 (0.003)** | 0.657 (0.064) |     0.096 (0.008) |
>
>
>
>
> > I am also skeptical of the finite sample estimator of the CMI regularizer.
>
> > Can the authors comment on the convergence of the proposed mutual information estimators with finite samples?
>
>
> We will add the following proposition to the paper, characterizing the finite sample convergence of the MI estimator.
>
> Proposition: Let $\\hat{I}\_n$ be the CMI estimator defined in Eq (7), estimated with IID samples $(x\_i, g\_i, \\tilde{y}\_i)_{i=1}^{n}$.
>
>  Under Assumptions 3.1 and 3.2, and that $\forall x_i \in \mathcal{B}: h_{\\phi}(f_{\\theta}(x_i)) = \\mathbb{P}(Y \\mid f_{\\theta}(x_i))$ and $ h_{\psi}(f_{\theta}(x_i)) =  \\mathbb{P}(Y \\mid f_{\\theta}(x_i), g_i)$, we have that, with probability $1 - \\delta$,
>
> $|\\hat{I}\_n - I(Y; G \\mid f(X))| \leq \sqrt{\frac{k \log (2/\delta)}{n}}  = O(\frac{1}{\sqrt{n}})$,
>
> where $k$ is a constant that depends only on $K$ and $|G|$.
>
> Further, combining this with Proposition 5.1, we have that
>
> $Suf_g(f) \\leq \\sqrt{\\frac{2}{K}} (\\hat{I}_n +  O(\\frac{1}{\\sqrt{n}}))^{1/2}$.
>
> We will include the full proof in the appendix in the revision, but it involves (1) showing the Lipschitzness of the mutual information in the total variation distance, (2) applying the Bretagnolle–Huber’s inequality on the two $P_{Y,G}$ distributions, and (3) plugging (2) into (1).

---

> > ### Comment · Reviewer_fbDW · 2025-08-03
> > **Thank you for the rebuttal**
> >
> > Thank you for your clarifications. Conditional on the new results being added to the paper and the convergence proof, I will upgrade my rating.

---

### Official Review · Reviewer_xUeM · 2025-07-04

**Clarity:** 3
**Significance:** 2
**Originality:** 2
**Rating:** 4
**Confidence:** 4

**Summary:**

The paper introduces a method for ensuring per-sensitive-group calibratedness in the presence of noise. Under the assumption that one knows the per-group transition probabilities for the noisy labels, the paper provides a two-step algorithm to calculate a calibration gap using a mutual information between the ground truth label and the group membership condition on the classifier output. In the first step, the mutual information is estimated, while in the second step, the model is trained using a regularizer informed by the mutual information bound. Empirical results are presented comparing the method with a baseline where the calibration is directly regularized.

**Questions:**

- Q1. On page 7, I think the transition matrix for Eq. (4) should probably be $T^{-1}$, not $T_{g_{i}}^{-1}$. No? Same for Eq. 6 and onwards, as none of these probability estimates have a conditioning on group $G$.
- Q2. Please discuss multi-calibration.

**Ethical Concerns:**

["NO or VERY MINOR ethics concerns only"]

**Final Justification:**

I have adjusted my score to a weak accept post rebuttal following additional experiments and discussions around multicalibration.

**Limitations:**

As also stated in the paper's limitation section (thanks for making this very clear), the assumption that we have the transition matrices as inputs to the algorithm is a rather strong one that limits the applicability of the algorithm as it is. I think the paper would have benefited greatly from an estimation step using a small, clean dataset (for example).

**Paper Formatting Concerns:**

None that I noted.

**Quality:**

2

**Strengths And Weaknesses:**

Overall, despite the interesting problem, this paper falls below the acceptance threshold for me.  I will enumerate **S**trengths and **W**eaknesses.
- **S1. The problem of fairness calibration under noise is relevant.** Given how much noise real-world data has and the fact that many risks of systematic discrimination are attributable to noisy data, this is a valuable problem to tackle.
- **W1. Lack of meaningful baselines.** The baseline discussed in the paper adds its regularizer to prior noise-robust methods to adapt them to the multi-group calibration problem with noisy data. This is useful and does show the versatility of the method, and allows judging the improvement that the use of CMI-REG and CMI-SUF brings to their respective methods. However, these are not competitive baselines for the problem. I understand that the authors claim that no prior work has tackled this exact problem, but it behooves them to construct reasonable baselines for comparison as a result. Here is my suggested baseline:

  Since the transition matrices are given to CMI-REG, why not transform the noisy data using the transition matrices $T^g$ and $T$ as a pre-processing step and then use any standard multicalibration algorithm?

  On the latter note, I find that lack of any discussion or reference to the (now) vast literature of multi-calibration is a miss.
- **W2. Novelty and contribution are not clear.** Methodically, the paper's approach can be summarized as adding a calibration regularizer with corrections to prediction rates due to noise, which is completely specified via transition matrices. Furthermore, the robustness to noise seems to be more a function of the base loss $\ell_{pred}$ in Eq. 2 than a contribution of the presented methods. Unfortunately, the experiments do not convince me otherwise, either due to small relative improvements, many of which fall within error margins.
- **W3. Uneven presentation.** The level of detail in the presentation is uneven. As an example, I was very surprised to find SUF-REG being presented in the experimental section (Line 254) in 2 lines with minimal details. The authors emphasize that this is a novel contribution. If so, why treat it as an afterthought? Especially given how competitive it can be with the main method, CMI-REG. This left me with more questions than answers, such as: If we can regularize sufficiency directly, what is the point of all the mutual information estimation?
  Similarly, backward and forward corrections in Lines 206 and 208 are presented with only a reference and no details.
- **W4. Empirical results are not very convincing.** I touched upon this briefly in W2. Looking at Figure 2, I cannot confirm that CMI-REG and SUF-REG indeed Pareto dominate in every case. In lsac+DMI and lsac+Peer that does not seem to be the case; also, Figure 3 in the appendix seems to be a similarly mixed set of results. Also, overall, the Pareto fronts are rather sparse. I understand that training and hyperparameter tuning may be difficult, but it's really hard to compare a single point to a full curve in a Pareto front setting.

---

> ### Author Rebuttal · Authors · 2025-07-31
>
> Thank you for the insightful review and constructive feedback!
>
> > W1. Lack of meaningful baselines. Here is my suggested baseline: Since the transition matrices are given to CMI-REG, why not transform the noisy data using the transition matrices $T^g$ and $T$ as a pre-processing step and then use any standard multicalibration algorithm?
>
> Thank you for the suggestion! There are two primary issues with the proposed baseline. First, the inverse of the transition matrix $(T^g)^{-1}$ is in general not row-stochastic, and may contain values outside of $[0, 1]$. Thus, it cannot not be directly used in a _pre-processing_  step to transform the observed labels into unbiased true labels. Second, applying multicalibration during test-time requires knowledge of the group attributes during inference. None of our methods or baselines require this assumption.
>
> Regardless, we have adapted the proposed baseline to our setup in the following way:
> - We apply a softmax to each row of $(T^g)^{-1}$ so it is row stochastic.
> - We randomly flip observed labels according to this matrix, and then train an ERM model on this processed data.
> - We apply Algorithm 1 from [1] for multicalibration, specifically using the implementation from [2], using the single attribute to define $\mathcal{C}$.
> - We vary all hyperparameters (including $\alpha$ and $\lambda$) using a random search, and select the optimal model using the same setting as Table 2 of the paper.
>
> We present results for AUROC and Sufficiency Gap below. We include only the Forward loss for our regularizers for brevity. We find that sufficiency gaps for this baseline remain substantially higher and AUROC is often worse.
>
>
> |                   |         |                 income  |          |                 adult  |           |              lsac     |         |                crime    |
> |-------------------|--------------:|------------------:|--------------:|------------------:|--------------:|------------------:|--------------:|------------------:|
> |                   |         AUROC |               Suf |         AUROC |               Suf |         AUROC |               Suf |         AUROC |               Suf |
> |           Forward | 0.801 (0.002) | **0.038 (0.002)** | 0.887 (0.001) | **0.024 (0.004)** | 0.436 (0.006) |     0.097 (0.001) | 0.788 (0.011) |     0.056 (0.010) |
> | Forward + Suf-Reg | 0.812 (0.002) |     0.059 (0.003) | 0.898 (0.000) |     0.032 (0.001) | 0.690 (0.011) |     0.091 (0.016) | 0.789 (0.006) |     0.053 (0.006) |
> | Forward + CMI-Reg | 0.801 (0.002) | **0.038 (0.001)** | 0.888 (0.002) | **0.024 (0.004)** | 0.704 (0.172) | **0.065 (0.070)** | 0.802 (0.015) | **0.046 (0.011)** |
> |         GroupPeer |             - |                 - | 0.898 (0.001) |     0.035 (0.004) | 0.664 (0.010) |     0.095 (0.023) |             - |                 - |
> |                Multicalibration | 0.561 (0.019) |     0.078 (0.010) | 0.787 (0.004) |     0.017 (0.001) | 0.500 (0.000) |     0.107 (0.007) | 	0.508 (0.013) |     0.104 (0.012) |
> |            Oracle | 0.854 (0.001) |     0.003 (0.000) | 0.919 (0.002) |     0.013 (0.001) | 0.838 (0.011) |     0.018 (0.009) | 0.825 (0.013) |     0.046 (0.010) |
>
>
>
> > W2. Novelty and contribution are not clear. Methodically, the paper's approach can be summarized as adding a calibration regularizer with corrections to prediction rates due to noise, which is completely specified via transition matrices. Unfortunately, the experiments do not convince me otherwise, either due to small relative improvements, many of which fall within error margins.
>
>
> We argue that our contributions extend beyond simply combining noise-robust losses with a heuristic regularizer. First, we propose and formalize this particular problem, which, to our knowledge, has not been studied in prior work. Second, we theoretically characterize this problem, showing necessary & sufficient conditions showing that label bias, even when classifiers are calibrated on noisy labels, inevitably induces a sufficiency gap on the true labels (Theorem 4.1 and Corollary. 4.1.1). Third, though our experimental gains are modest, they do seem to be consistent, and statistically significant in many cases.
>
>
> > W3. Uneven presentation. As an example, I was very surprised to find SUF-REG being presented in the experimental section (Line 254) in 2 lines with minimal details. Similarly, backward and forward corrections in Lines 206 and 208 are presented with only a reference and no details.
>
> Due to space issues, we were not able to describe SUF-REG thoroughly in the main paper and had to relegate it to the appendix. We focused primarily on CMI-REG because it performed marginally better, and because the techniques used in CMI-REG (e.g. linear heads for posterior estimation) are common across the two methods. The backward correction used in L206 and L208 are applications of a well-known result from prior work [3], and so we did not describe it in detail. We will provide more exposition to this in the revision.
>
>
> > W4. Empirical results are not very convincing. Looking at Figure 2, I cannot confirm that CMI-REG and SUF-REG indeed Pareto dominate in every case. Also, overall, the Pareto fronts are rather sparse.
>
> We note that expecting our regularizer to improve every single base loss is a lofty goal that may not be realistic, as some losses may have optimization landscapes that naturally work better with our regularizer. We note that overall, either CMI-REG or SUF-REG achieves the lowest sufficiency gap in 8/10 experimental settings (Tables 3 and 4 in the appendix).
>
> Regarding the sparsity of the Pareto plots, we emphasize that all methods were given the same budget of 20 hyperparameter from a random search, from a fairly wide range grid shown in Appendix B.3. The reason why some Pareto curves collapse to a single point is because there is no observed trade-off between fairness and performance in those cases. In general, such a trade-off may not exist, e.g. the group Bayes optimal score with respect to true labels has maximum performance and zero sufficiency gap.
>
>
> > Q1. On page 7, I think the transition matrix for Eq. (4) should probably be $T^{-1}$, not $(T_g)^{-1}$. No? Same for Eq. 6 and onwards, as none of these probability estimates have a conditioning on group.
>
> In our class-attribute-conditioned noise model (Assumption 3.1), the backward correction always involves $T_g$ because noise depends on $(Y, G)$, which leads to $\\mathbb{E}_{\\tilde Y \\mid Y = y,\\, G = g} [(T_g^{-1})\_{\\tilde{Y},\\cdot}] = e\_y^{\\top}$. Using the marginal matrix $T$ would result in a biased estimator, as it does not correspond to the actual probability of corruption for that sample.
>
> > On the latter note, I find that lack of any discussion or reference to the (now) vast literature of multi-calibration is a miss.
>
> > Q2. Please discuss multi-calibration.
>
>
> We will add a paragraph on multicalibration to the related works in Section 2.2, and add the above experiment with the multicalibration baseline to the main paper. Please let us know if this is satisfactory!
>
>
> > As also stated in the paper's limitation section (thanks for making this very clear), the assumption that we have the transition matrices as inputs to the algorithm is a rather strong one that limits the applicability of the algorithm as it is. I think the paper would have benefited greatly from an estimation step using a small, clean dataset (for example).
>
> Thank you for the suggestion! We note that we do experiment with using Confident Learning [4] to estimate the transition matrix (only from noisy data), though this generally results in worse performance than using the true transition matrix. We emphasize that as better transition matrix estimations methods are developed, our method will improve in this setting as well.
>
> If we were instead given $n$ iid samples with labels for whether they are mislabeled or not as the reviewer suggests, we could then use the empirical estimator for each element of $T$. We have theoretically characterized the behavior of this estimator, and its impact on sufficiency gap estimates, in Appendix A.5.
>
>
>
> [1] Multicalibration: Calibration for the (Computationally-Identifiable) Masses. ICML 2018.
>
> [2] When is Multicalibration Post-Processing Necessary? NeurIPS 2024.
>
> [3] Making Deep Neural Networks Robust to Label Noise: a Loss Correction Approach. CVPR 2017.
>
> [4] Confident Learning: Estimating Uncertainty in Dataset Labels. JAIR.

---

> > ### Comment · Reviewer_xUeM · 2025-08-04
> >
> > Thank you for the rebuttal. In particular, thanks for implementing the baseline. While I appreciate the effort, some things that stood out to me about the results:
> > a) Most of your multicalibration baselines are nearly random classifiers (AUCs around 50%). I am skeptical this is not a bug.
> > b) On adult, you highlight 0.024 (0.004) as your best achieved gap when multicalibration achieves 0.017 (0.001). Seems like a typo, but important to note nevertheless.
> >
> > Thanks for the clarification regarding other concerns. To be clear, I have seen both the theorem and corollary proofs, and while I appreciate the formalization and the proofs, I do not think these results hold up as major contributions to your work. I think the problem setup and experimental work are by far more important. I want to stress that I find no issue with not having as much theory. My point is that your contributions are mostly to the formulation and evaluation of the problem. And in that department, the paper, in my humble opinion, is lacking.
> >
> > I will adjust my score to a weak accept. I think the best way to improve this paper is to really dig into the multicalibration angle. Explain why or why not it does not work. Put the theory to good use there if you can. It will make for a better contribution.
> >
> > I wish you the best.

---

### Note · Authors · 2025-08-15

We thank all reviewers again for their valuable feedback! We are glad that reviewers recognized the importance of studying group sufficiency under label noise (xUeM, HAo2), the value of our theoretical characterization (fbDW, NBv5), and the breadth of the experiments (HAo2, fbDW, NBv5). We appreciate that all reviewers acknowledged and responded to our rebuttals, engaging in fruitful discussions.

As a result of these discussions, we have added the following to the revision:

**Multicalibration baseline + discussion**: Following the suggestion from Reviewer xUeM, we implemented a multicalibration baseline with noise-aware preprocessing, inserted results into the main tables/figures, added a related-work paragraph (Sec. 2.2) and a discussion of why multicalibration is mismatched to our setting (Sec. 8).

**Additional noise rates + "Hardest setting" results**: Following the suggestion from Reviewer HAo2, we have added results at two additional noise rates (0.5 and 0.6) and with side-by-side comparisons of true vs. estimated transition matrices for both CMI-REG and SUF-REG. In addition, following the suggestion from Reviewer fbDW, we have added results for the setting where the transition matrix is estimated and noisy Brier is used for selection.

**Additional dataset**: Following the suggestion from Reviewer NBv5, we have added an additional tabular dataset with real-world noise for predicting student exam pass/fail.

**Clarification of Eq. (4)**: Following the discussion of Equation (4) with reviewer HAo2, we have now clarified its derivation in the main text and in the proof of Lemma A.5, adding particular detail to why $T_g$ is used in the estimate of $H(Y | f(X))$.

**Finite-sample guarantees for the MI estimator**: Following the suggestion from Reviewer fbDW, we have added a new proposition with a finite-sample convergence bound for our conditional mutual information estimator to Appendix A.


These changes have substantively strengthened the paper. We believe that all reviewers now lean positively toward acceptance; in particular, Reviewer xUeM is currently at a borderline reject but indicated that they would "adjust [their] score to a weak accept". We again thank all reviewers for their constructive feedback and thoughtful engagement!

---

### Decision · Program_Chairs · 2025-09-17

**Decision:**

Accept (poster)

**Comment:**

The paper tackles the important problem of achieving group sufficiency (equal calibration across demographic groups) under label bias, where observed labels differ systematically from true labels at different rates across groups. The authors provide theoretical analysis showing that enforcing fairness with respect to biased labels necessarily results in group miscalibration with respect to true labels, and propose a conditional mutual information-based regularizer (CMI-REG) to minimize an upper bound on the sufficiency gap. They evaluate their approach across eight datasets with both synthetic and real label noise, demonstrating consistent improvements in sufficiency gap reduction of up to 9.1% without significant accuracy loss.

The reviewers and authors engaged in substantive technical discussions that strengthened the paper considerably. Reviewer xUeM raised concerns about the lack of meaningful baselines and suggested implementing a multicalibration baseline, which the authors successfully added along with related work discussion. Reviewer fbDW questioned the method's performance when transition matrices must be estimated rather than known, prompting the authors to provide extensive additional experimental results for this challenging setting. Reviewer HAo2 identified confusion in the derivation of Equation 4 regarding the entropy term conditioning, leading to detailed mathematical clarification from the authors about why group-specific transition matrices Tg are necessary for unbiased estimation despite the entropy not explicitly depending on group membership. Reviewer NBv5 suggested additional datasets and experimental comparisons, resulting in the inclusion of results on real-world biased label datasets. Through this iterative process, all reviewers moved toward acceptance, with the final paper incorporating multicalibration baselines, additional noise levels, finite-sample convergence guarantees, and clearer mathematical exposition.